# Lifelong single-cell profiling of cranial neural crest diversification in zebrafish

Peter Fabian [1,3], Kuo-Chang Tseng [1,3], Mathi Thiruppathy [1,3], Claire Arata [1,3], Hung-Jhen Chen[1], Joanna Smeeton [1,2], Nellie Nelson [1] & J. Gage Crump [1✉]

The cranial neural crest generates a huge diversity of derivatives, including the bulk of connective and skeletal tissues of the vertebrate head. How neural crest cells acquire such extraordinary lineage potential remains unresolved. By integrating single-cell transcriptome and chromatin accessibility profiles of cranial neural crest-derived cells across the zebrafish lifetime, we observe progressive and region-specific establishment of enhancer accessibility for distinct fates. Neural crest-derived cells rapidly diversify into specialized progenitors, including multipotent skeletal progenitors, stromal cells with a regenerative signature, fibroblasts with a unique metabolic signature linked to skeletal integrity, and gill-specific progenitors generating cell types for respiration. By retrogradely mapping the emergence of lineage-specific chromatin accessibility, we identify a wealth of candidate lineage-priming factors, including a Gata3 regulatory circuit for respiratory cell fates. Rather than multilineage potential being established during cranial neural crest specification, our findings support progressive and region-specific chromatin remodeling underlying acquisition of diverse potential.

[1] Eli and Edythe Broad California Institute for Regenerative Medicine Center for Regenerative Medicine and Stem Cell Research, Department of Stem Cell Biology and Regenerative Medicine, University of Southern California Keck School of Medicine, Los Angeles, CA 90033, USA. [2] Department of Rehabilitation and Regenerative Medicine, Columbia University Irving Medical Center, Columbia University, New York, NY 10032, USA. [3] These authors contributed equally: Peter Fabian, Kuo-Chang Tseng, Mathi Thiruppathy, Claire Arata. ✉email: gcrump@usc.edu

Cranial neural crest-derived cells (CNCCs) are a vertebrate-specific population, often referred to as the fourth germ layer, that have extraordinary potential to form diverse cell types. In addition to pigment cells and the peripheral nervous system, CNCCs form the ectomesenchyme that populates the pharyngeal arches and gives rise to much of the skeleton and connective tissue of the jaws and face[1]. Posterior arch CNCCs contribute to a distinct set of organs, including the thymus, parathyroid, and cardiac outflow tract, and in fishes cell types important for respiration, including specialized endothelial-like pillar cells that promote gas exchange[2]. In zebrafish, teeth develop from CNCCs of the most posterior seventh arch.

The extent to which diverse lineage potential is pre-patterned in CNCCs, or acquired after their migration to the arches, has been investigated for over a century through labeling, grafting, and extirpation experiments[3]. Individual avian CNCCs can generate multiple types of derivatives in vitro, including ectomesenchyme and neuroglial cells, suggesting multilineage potential is an intrinsic property[4]. In addition, grafting of avian CNCCs between arches showed that they retain a pre-pattern of which skeletal elements to make, such as the beak[5]. On the other hand, grafting of endodermal epithelia can induce supernumerary skeletal elements of morphologies corresponding to the location from which epithelia are taken, suggesting an important role of external cues in CNCC fate[6]. Further, grafted trunk neural crest cells, which normally do not make mesenchymal derivatives, can contribute to the facial skeleton following extended culture[7] or misexpression of key transcription factors[8]. And in skate, mesodermal cells can contribute to gill cartilage, a classically considered CNCC-derived structure, suggesting that, rather than cell potential being a unique intrinsic property of CNCCs, extrinsic cues from the local arch environment may induce similar cell fates in mesenchyme of diverse origins in certain contexts[9].

Here we take a genomics approach in zebrafish to understand when enhancers linked to diverse CNCC fates first gain accessibility. In so doing, we find that chromatin accessibility underlying multilineage potential is largely gained after CNCC migration, arguing against an epigenetic pre-pattern in premigratory CNCCs for diverse fate potential.

## Results

**Single-cell atlas of CNCC derivatives across the zebrafish lifetime.** To understand the emergence and diversification of CNCC lineages across the lifetime of a vertebrate, we constructed a longitudinal single-cell atlas of gene expression and chromatin accessibility of zebrafish CNCC derivatives. We permanently labeled CNCCs using *Sox10:Cre; actab2:loxP-BFP-STOP-loxP-dsRed* (*Sox10>dsRed*) fish (Fig. 1a), in which genetic recombination indelibly labels CNCCs shortly after their specification at 10 h post-fertilization (hpf)[10]. Previous single-cell analyses of CNCCs in zebrafish[11], chick[12], and mouse[13–15], and in vitro human CNCC-like cells[16], had focused on CNCC establishment, migration, and early fate choices between the neuroglial, pigment, and ectomesenchyme lineages. Here we investigate cellular diversity and lineage progression of CNCC ectomesenchyme across embryonic (1.5 and 2 days post-fertilization (dpf)), larval (3 and 5 dpf), juvenile (14 and 60 dpf), and adult (150–210 dpf) stages. After fluorescence-activated cell sorting (FACS) of *Sox10>dsRed*+ cells from dissected heads, we performed single-cell RNA sequencing (scRNAseq) and single-nuclei assay for transposase accessible chromatin sequencing (snATACseq) at each stage using the 10X Genomics Chromium platform and paired-end Illumina next-generating sequencing (Fig. 1b). After filtering for quality, we obtained 58,075 cells with a median of 866 genes per cell for scRNAseq, and 88,177 cells with a median of

10,449 fragments per cell for snATACseq (Supplementary Dataset 1). To better resolve snATACseq data, we used the SnapATAC package[17], which integrates snATACseq with scRNAseq data to create pseudo-multiome datasets.

Analysis of CNCC cell clusters across all stages using UMAP dimensionality reduction recovered most known CNCC derivatives, including Schwann cells (glia), several neuronal subtypes, pigment cells, and diverse mesenchymal cell types (Figs. S1–8, Supplementary Dataset 2, and Supplementary Table 1). We also recovered otic placode and epithelial cells, likely reflecting additional non-CNCC expression of *Sox10:Cre*[10], and blood lineage cells, likely due to autofluorescence. Similar clusters were recovered using both scRNAseq and SnapATAC data. We then re-clustered the CNCC ectomesenchyme sub-population across stages, as this makes the most substantial and diverse cell contributions in the head. To confirm ectomesenchyme identity at 1.5 dpf, we also performed scRNAseq analysis of cells double-positive for the CNCC transgene *sox10:dsRed* and the ectomesenchyme transgene *fli1a:eGFP*. Co-clustering showed high concordance between *sox10:dsRed*+; *fli1a:eGFP*+ ectomesenchyme and the *Sox10>dsRed*+ ectomesenchyme subset, and among *Sox10>dsRed*+ ectomesenchyme scRNAseq subsets at all 7 stages (Fig. S8).

At the adult stage, we recovered 17 distinct clusters using scRNAseq that corresponded to 16 clusters using SnapATAC; these were largely associated with the jaw skeleton or gills (Fig. 1c–e). Skeletal derivatives include bone, cartilage, teeth, and a population with properties of periosteum, tendon, and ligament. Gills are composed of primary filaments containing cartilage rods and primary veins surrounded by a tunica media, and numerous secondary filaments housing endothelial-like pillar cells that promote gas exchange. We recovered gill chondrocytes, pillar cells, tunica media cells, and putative gill progenitors, with gill cartilage being distinct from hyaline cartilage of the jaw at both the transcriptional and chromatin accessibility levels. We also recovered smooth muscle, perivascular, and stromal cells (see Supplementary Dataset 2 and Supplementary Table 1 for cluster marker genes and Figs. S9–10 for in situ validation).

In addition to skeletal and gill populations, we recovered a distinct type of fibroblast enriched for the cell adhesion molecule *chl1a* and *wnt5a*. Strikingly, these fibroblasts are enriched for genes encoding enzymes for all steps of phenylalanine and tyrosine breakdown (Fig. 1f, Fig. S11). In situ hybridization for two of these genes (*hpdb* and *pah*) reveals that these fibroblasts are in the dermis between the skin epidermis and *runx2b*+/*sp7*+ osteoblast lineage cells (Fig. 1g, h). Our findings, therefore, show distinct types of cells in the jaws versus gills, as well as a specialized type of CNCC-derived fibroblast between the skeleton and skin.

**Progressive emergence of CNCC derivatives and region-specific progenitors.** We next sought to understand whether the diverse cell types of the jaws and gills arise from common or distinct types of progenitors. First, we applied the STITCH algorithm[18] to scRNAseq and snATACseq datasets to connect individual stages; the force-direct layout of STITCH allows visualization of both cell relatedness and developmental trajectories in 2D space (Fig. 2a, b). We then used in situ validation to assign regions of the scRNAseq STITCH plot to major cell types, and transferred identities to the snATACseq STITCH plot based on chromatic accessibility across gene bodies (i.e., pseudo-transcriptome) (Figs. S9, S10). As early as 3 dpf (particularly apparent for snATACseq), we observe partitioning of CNCCs into skeletogenic versus gill lineages. Skeletal-associated regions include *hyal4*+ perichondrium, periosteum, tendon/ligament cells, chondrocytes, and osteoblasts (Fig. S9), and

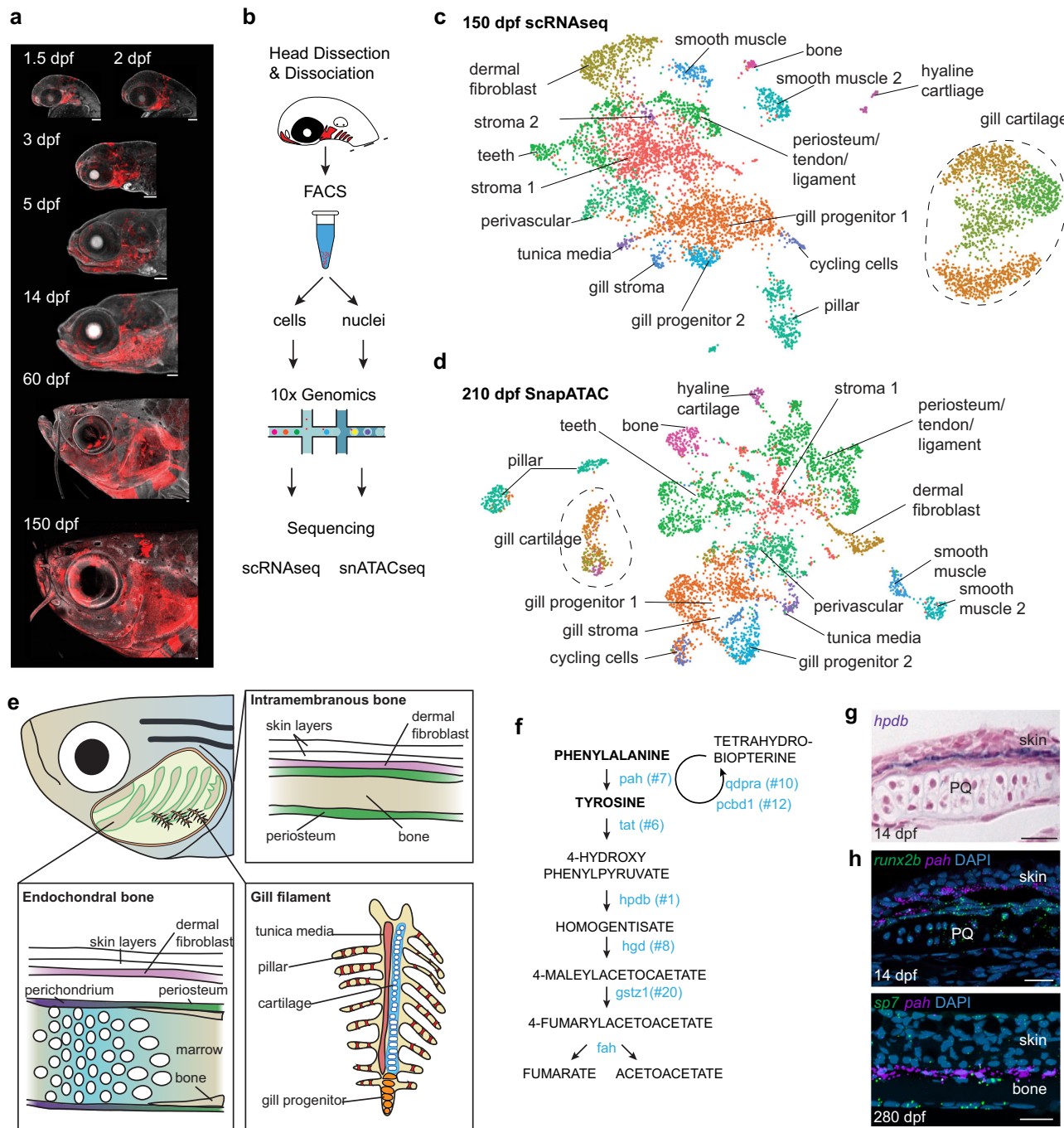

**Fig. 1 Single-cell transcriptomes and chromatin accessibility of CNCCs across the zebrafish lifetime. a** *Sox10:Cre; actab2:loxP-BFP-STOP-loxP-dsRed* labeling of CNCCs (red) in zebrafish heads across 7 stages. **b** Scheme of cell or nuclei dissociation, fluorescence-activated cell sorting (FACS), and processing on the 10X Genomics platform for sequencing. **c, d** UMAPs of scRNAseq and SnapATAC datasets at adult stages. **e** Diagram of an adult zebrafish head cut out to show major cell types of the interior skeletal elements and gill system. **f** Pathway for Phe and Tyr breakdown. 7 of 8 genes encoding catabolic enzymes are in the top 20 selectively enriched genes for the dermal fibroblast cluster (numbers show rank). **g** Section colorimetric in situ hybridization for *hpdb* RNA shows expression in the dermis between the skin and palatoquadrate (PQ) endochondral bone. **h** Section RNAscope in situ hybridizations show *pah* expression in dermal fibroblasts between the *runx2b*+ periosteum of PQ and skin at 14 dpf, and between *sp7*+ osteoblasts of intramembranous bone and skin at 280 dpf. DAPI labels nuclei in blue. Scale bars, 100 μm (**a**), 20 μm (**g, h**).

gill-associated clusters include an *fgf10b*+ gill progenitor population, gill chondrocytes, pillar cells, and tunica media cells (Fig. S10). Starting at 3 dpf, dermal fibroblasts form a separate branch in both scRNAseq and snATACseq STITCH plots, and *cxcl12a*+ stromal cells and teeth are closely associated with less differentiated mesenchyme in the center of the plots (Fig. 2a, b; Figs. S9, S11).

By creating an index for ectomesenchyme-enriched gene expression at 1.5 dpf, a stage preceding the onset of differentiation, we found no evidence for retention of ectomesenchyme identity at later stages, as shown by aggregated ectomesenchyme gene expression and the early ectomesenchyme marker *nr2f5*[19] (Fig. S12). Although formation of CNCC ectomesenchyme involves a transient reacquisition of the pluripotency network[16],

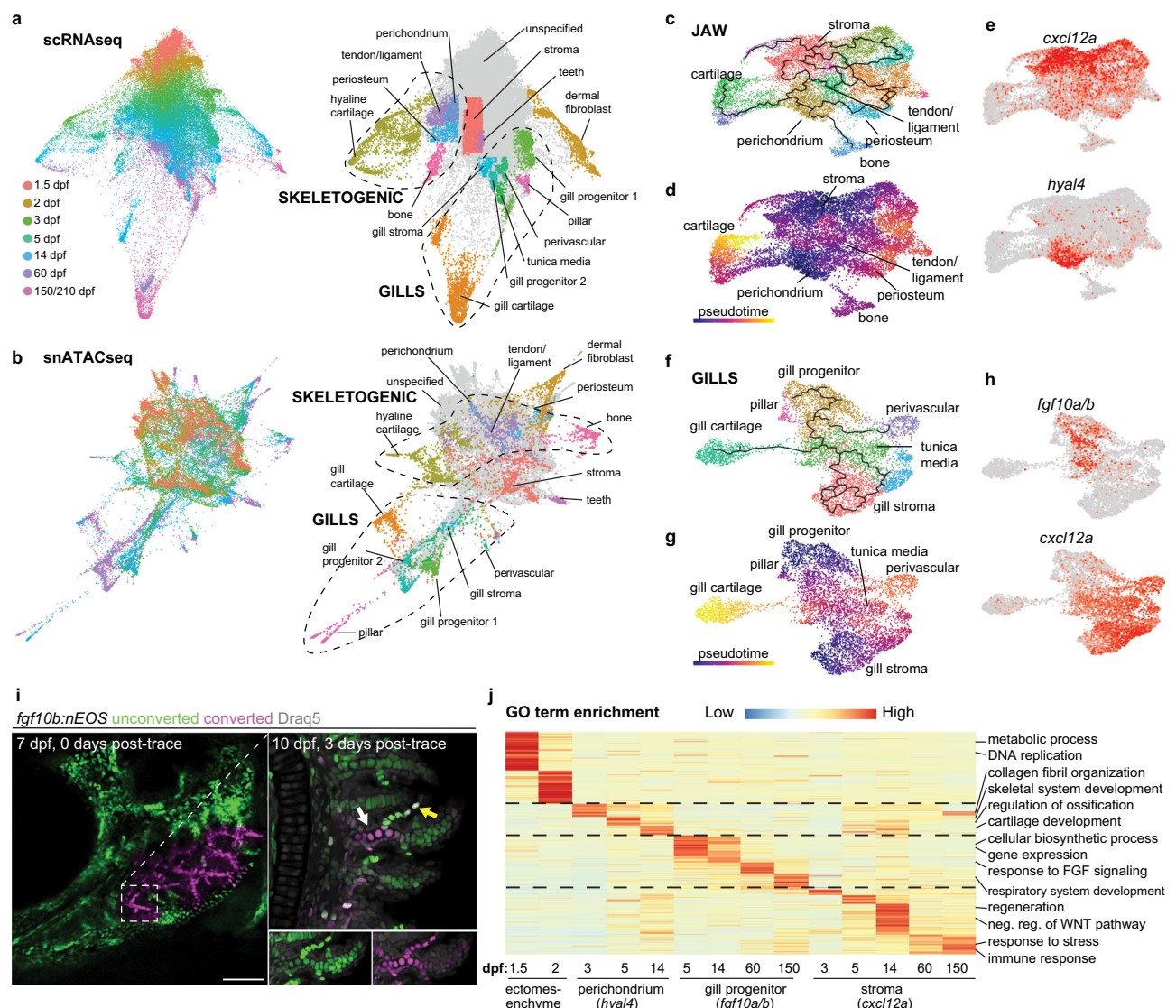

**Fig. 2 Progressive emergence of region-specific lineage programs. a, b** STITCH plots connect individual scRNAseq and snATACseq datasets across the zebrafish lifetime. Cell type annotations show divergence of CNCC ectomesenchyme into skeletogenic, gill, dermal fibroblast, and stromal branches. **c–e** Pseudotime analysis using Monocle3 of the jaw skeletogenic subset (combined 5 and 14 dpf) shows a *cxcl12a+* stromal branch, and a *hyal4+* branch connected to cartilage, bone, periosteum, and tendon and ligament. **f–h** Pseudotime analysis of gill subsets (combined 5 and 14 dpf) shows a *cxcl12a+* stromal branch, and a *fgf10a/b+* branch connected to gill pillar, tunica media, and perivascular cells, as well as a distinct type of gill cartilage. **i** Following UV-mediated photoconversion of *fgf10b:nEOS* from green to magenta in a subset of filaments, re-imaging 3 days later revealed contribution of converted cells to gill chondrocytes (white arrow) and pillar cells (yellow arrow, white signal reflects mixture of previously converted magenta and continued expression of new unconverted green *fgf10b:nEOS*) (*n* = 3). Draq5 labels nuclei in gray. At 10 dpf, below insets show the region designated by the arrows for individual green and magenta channels with Draq5. **j** Gene ontology (GO) terms of biological processes for the respective clusters at the stages indicated. Heatmap reflects the negative log of the adjusted *p* value of one-sided Fisher's exact test. Ectomesenchyme is defined as the average of all cells at 1.5 and 2 dpf stages. Scale bar, 100 μm.

we also did not observe expression of pluripotency genes *pou5f3* (*oct4*), *sox2*, *nanog*, and *klf4* at any stage of post-migratory ectomesenchyme, although we cannot rule out low expression of some or all of these genes. One exception was *lin28aa* that displays broad expression at 1.5 dpf and is rapidly extinguished by 2 dpf (Fig. S12). Rather than maintenance of a multipotent ectomesenchyme population, our data point to progressive emergence of specialized *hyal4+* perichondrium, *cxcl12a+* stromal, and *fgf10a/b+* gill populations at 3 dpf and beyond (Fig. S12).

To further dissect region-specific lineages, we used Monocle3[20] on scRNAseq datasets to construct pseudotime trajectories of gill (*hoxb3a+/gata3+* cells) versus jaw (remaining cells after

removing dermal fibroblasts and teeth) mesenchyme at 5–14 dpf (Fig. S13). For jaw clusters, cell distribution from 5 to 14 dpf suggested two distinct lineages: one involving chemokine-expressing stromal cells (*cxcl12a+/ccl25b+*) and a second emanating from *hyal4+* cells that express lower levels of *cxcl12a* (Fig. 2c–e, Fig. S13). In situ hybridization at 14 dpf revealed broad mesenchymal expression of *cxcl12a*, and expression of *hyal4* in perichondrium in a largely complimentary pattern to *postnb* and *col10a1a* expression in periosteum (Fig. S9). Branches from *hyal4+* perichondrium led to periosteum, tendon and ligament cells, chondrocytes, and osteoblasts, consistent with studies showing perichondrium to be the precursor of the periosteum in endochondral bones[21,22].

For gill clusters, cell distribution from 5 to 14 dpf revealed two primary trajectories (Fig. 2f–h, Fig. S13). In the first, *cxcl12a* +/*ccl25b*+ stromal cells give rise to mesenchyme associated with retinoic acid metabolism (*aldh1a2*+/*rdh10a*+), with in situ hybridization revealing these cell types restricted to the base of secondary filaments (Fig. S10). In the second, *fgf10a*+ cells are connected to *fgf10b*+ cells, which then diverge into gill cartilage, pillar cells, and perivascular cells. To test whether *fgf10b*+ cells are progenitors for specialized gill subtypes, we used CRISPR/Cas9 to insert a photoconvertible nuclear EOS protein into the endogenous *fgf10b* locus. We found *fgf10b*:nEOS to be robustly expressed in the forming gills, with expression becoming progressively restricted to the tips of gill filaments from 5 dpf to 2 years of age, similar to endogenous *fgf10b* expression (Figs. S10, S14). We then used UV light to convert *fgf10b*:nEOS fluorescence from green to red in a subset of filaments at 7 dpf and observed contribution to gill chondrocytes and pillar cells 3 days later (Fig. 2i). Similar results were seen in adult gill filaments (Fig. S14). These data support *fgf10b*+ cells being progenitors for gill cartilage and pillar cells from larval through adult stages; the origins of tunica media and perivascular populations remain unresolved.

We next examined how gene usage and chromatin accessibility in CNCC mesenchyme changes during aging to meet the differing demands of the rapidly growing embryo versus homeostatic adult (Fig. 2j, Figs. S15–17, Supplementary Dataset 3). We observed the largest enrichment of chromatin remodelers in the ectomesenchyme population at 1.5 and 2 dpf, followed by tissue-specific progenitors at 3 and 5 dpf, suggestive of large-scale remodeling of lineage-specific enhancers at these stages. Gene ontogeny (GO) analysis of ectomesenchyme at 1.5 and 2 dpf revealed terms linked to cell division and metabolism, consistent with early expansion of this population. We also find enrichment of transcription factors for early ectomesenchyme (e.g., *dlx2a*, *twist1a*, *nr2f6b*) and arch patterning (e.g., *pou3f3b*, *hand2*), as well as transcription factor binding motifs for several types of nuclear receptors, in accordance with known roles of Nr2f members in ectomesenchyme development[19]. The *hyal4*+ population contains skeletal-associated terms (collagen fibril organization, skeletal system development, regulation of ossification, cartilage development), consistent with its position as a common progenitor for cartilage, tendon, ligament, and bone in pseudotime analysis. The *hyal4*+ population is enriched for transcription factors implicated in perichondrium biology (*mafa*, *foxp2*, *foxp4*)[23,24] and cartilage formation (*barx1*, *sox6*, *emx2*)[25–27], as well as motifs for Bmp signaling (SMAD) and transcription factors (NFAT, RUNX) known to regulate cartilage and bone[28]. For *fgf10a/b*+ gill progenitors, we recover terms for general growth (e.g., translation, cellular biosynthetic process), response to Fgf signaling, and respiratory system development, consistent with lineage tracing showing *fgf10b*:nEOS-labeled cells giving rise to gill respiratory cell types through adult stages. We also observe enrichment of *gata2a*, *gata3*, and GATA motif accessibility, suggesting important roles of Gata factors in gill-specific lineages.

In contrast to *hyal4*+ and *fgf10a/b*+ populations that display hallmarks of progenitors, the *cxcl12a*+ stromal population is associated with terms for regeneration, response to injury and wounding, negative regulation of the Wnt signaling pathway, and, particularly at adult stages, response to stress and modulation of the immune response. This population is enriched for *osr1*, early response genes of the Fos/Jun family, C/EBP family members implicated in response to inflammation[29], and *egr1* that has recently been linked to injury-induced regenerative responses across the animal kingdom[30]. Recovery of motifs for STAT and C/EBP also point to immune system interactions. Together, our

analysis of CNCC trajectories reveals the progressive emergence of specialized progenitors in distinct regions such as the jaws and gills.

**Highly resolved embryonic spatial expression domains from integrated datasets.** Despite progress in understanding developmental factors that induce regional gene signatures in embryonic arch CNCCs, we still have an incomplete understanding of how these regional signatures influence later cell fate acquisition. As a first step in connecting early patterning to later region-specific differentiation, we examined the ability of integrated transcriptomic and chromatin accessibility datasets to predict the expression patterns of potential ectomesenchyme patterning genes at 1.5 dpf, a stage before overt cell type differentiation. Compared to scRNAseq (Fig. 3a) or snATACseq alone (Fig. S18), SnapATAC pseudo-multiome analysis (Fig. 3b) was better able to separate CNCCs along the major positional axes, including the dorsal-ventral axis and the anterior-posterior axis (frontonasal, mandibular (arch 1), hyoid (arch 2), gill-bearing (arch 3–6), and tooth-bearing (arch 7)).

Analysis of the predicted SnapATAC expression of known region-specific genes—*pou3f3b* (dorsal arches 1 and 2), *dlx5a* (intermediate arches), *hand2* (ventral arches), *meis2b* (arch 7), and *pitx1* (oral mandibular)[25,31,32]—revealed tight correlation to reported expression, including zebrafish-specific overlap of *dlx5a* and *hand2* in the mandibular arch (Fig. 3d). We also identified a previously unappreciated oral-aboral axis of the mandibular arch in zebrafish, marked by *pitx1* and *nr5a2*, respectively, which we validated by in situ hybridization for *nr5a2* (Fig. 3e). Re-examination of genes identified from previous bulk RNA sequencing of zebrafish arches further revealed strong correlation of SnapATAC domains with reported expression for 23 of 27 genes (Fig. S19), with SnapATAC suggesting frontonasal and tooth-domain expression for two genes previously annotated as false positives[25]. We also observed correlation of the transcription factor binding motifs enriched in cluster-specific accessible chromatin with the activities of transcription factors of the same family, including POU3F3, MEIS2, HAND, DLX5, PITX1, and NR5A2 (Fig. 3c, d). This approach shows the power of integrated scRNAseq and snATACseq data to predict the spatial expression domains of the vast majority of CNCC ectomesenchyme genes at pharyngeal arch stages, as well to uncover domain-specific transcription factor binding motifs such as NR5A2 (Supplementary Dataset 2).

**Chromatin accessibility predicts cell type competency in early arches.** We next sought to understand how the establishment of cell fate competency is linked to the earlier activity of arch patterning genes. To do so, we first computed unique patterns of chromatin accessibility (peaks) for each cell cluster at 14 dpf (Fig. 4a, Supplementary Dataset 4). Modules of the top enriched peaks for each cell type were then mapped onto UMAP projections of SnapATAC data at 1.5, 2, 3, and 5 dpf (Fig. S20). To understand when cluster-specific peaks become established, as well as cluster relatedness, we developed the bioinformatics pipeline Constellations to retrogradely map accessible chromatin regions related to differentiated cell types onto earlier stages preceding overt differentiation. First, we calculated whether projections of cluster-specific peak modules are skewed toward particular regions of UMAP space at each earlier time-point, suggesting establishment of cluster-specific chromatin accessibility (a proxy for cell type competency). We then computed the relatedness of peak module projections in two dimensions for each mapped cluster at each stage (Fig. 4b). Analysis of cell competency trajectories showed that cell types can be grouped

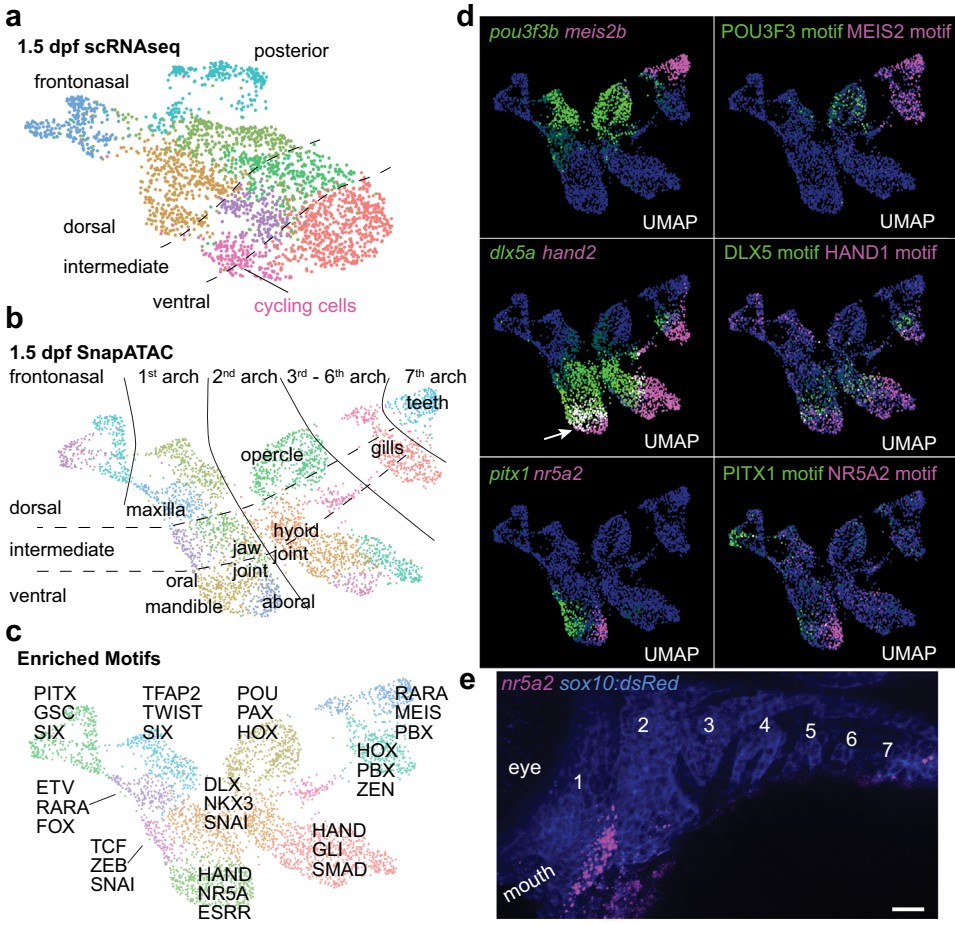

**Fig. 3 Highly resolved embryonic spatial expression domains from integrated datasets. a**, **b**, UMAPs at 1.5 dpf generated by scRNAseq versus integration of scRNAseq and snATACseq datasets using SnapATAC. SnapATAC outperforms scRNAseq in resolving dorsoventral (vertical), anterior-posterior (horizontal), and major arch landmarks including a previously unappreciated oral-aboral axis in zebrafish. **c** Select enriched transcription factor binding motifs for each cluster. **d** Gene body activities for transcription factors and their corresponding DNA-binding motifs reveal tight correspondence with published expression patterns, including zebrafish-specific overlap of *dlx5a* and *hand2* in the mandibular arch (arrow). **e** Fluorescent in situ hybridization shows restricted expression of *nr5a2* in the aboral domain of the mandibular arch as predicted by SnapATAC. *sox10:dsRed* labels CNCCs of the arches in blue (numbered, anti-dsRed antibody stain). Scale bar, 20 μm.

into five main classes: skeletogenic cells (including *hyal4*+ perichondral and *postnb*+ periosteal cells), stromal cells, dermal fibroblasts, gill cell types, and cartilage. These predicted cell groupings by Constellations analysis are robust to changes in resolution, which alters the number of predicted clusters at 14 dpf (Fig. S21a–e), as well as to inclusion of later stages (60 and 210 dpf) (Fig. S21f). Constellations analysis also reveals a temporal order of cell type competency establishment, with unique chromatin accessibility for cartilage and dermal fibroblast lineages emerging at 1.5 dpf; bone and perichondrium at 2 dpf; and periosteum, tendon/ligament, gill progenitors, and pillar cells at 3 dpf (Fig. 4c). This analysis suggests that chromatin accessibility prefiguring diverse CNCC cell types is progressively established rather than being inherited from earlier multipotent CNCCs.

**Constellations analysis reveals candidate transcription factors for lineage priming.** The power of the Constellations analysis is in identifying some of the earliest potential transcription factors for establishing cell type competency, based on expression of transcription factor genes and co-enrichment of their binding motifs in the same clusters. We identified 287 transcription factor expression/motif pairs showing co-enrichment (Fig. S22, Supplementary Dataset 5). The FOXC1 motif and *foxc1b* gene body activity were co-enriched in

the cartilage trajectory, and LEF1/*lef1* in the dermal fibroblast trajectory (Fig. 5a). Projection of FOX motifs and merged Fox gene activity (*foxc1a*, *foxc1b*, *foxf1*, *foxf2a*, *foxf2b*) and LEF1/*lef1* onto SnapATAC UMAPs at 1.5 dpf reveals close correlation to mapping of the 14 dpf peak modules for cartilage and dermal fibroblasts, respectively, at this stage (Fig. 5b, c). Both the Fox module and the cartilage-specific peaks also correlate closely with the known fate map of cartilage precursors in the arches[33] (Fig. 5d, e). This confirms genetic evidence for roles of Foxc1 and Foxf1/2 in cartilage formation in zebrafish and mouse[34,35], and more specifically Foxc1 in establishing accessibility of cartilage enhancers in the developing face[28]. It also raises the possibility that Wnt signaling, mediated in part by Lef1, may play a role in early dermal fibroblast specification, consistent with enrichment of *wnt5a* in this population (Fig. S11).

We also sought to understand how CNCCs in the gill region generate respiratory-type cells different from the skeletal cells in the jaw region. We find GATA3/*gata3* to be highly enriched in gill populations, with SnapATAC UMAP projections of GATA3 motif and *gata3* gene body activity at 5 dpf correlating with mapping of 14 dpf gill progenitor peaks (Fig. 5f). The enrichment of ETS2/*ets2*, which plays a role in endothelial development[36], in the gill pillar trajectory is consistent with ETS factors driving a mesenchyme-to-endothelia transition during formation of these vascular cells. In contrast, skeletogenic trajectories are uniquely

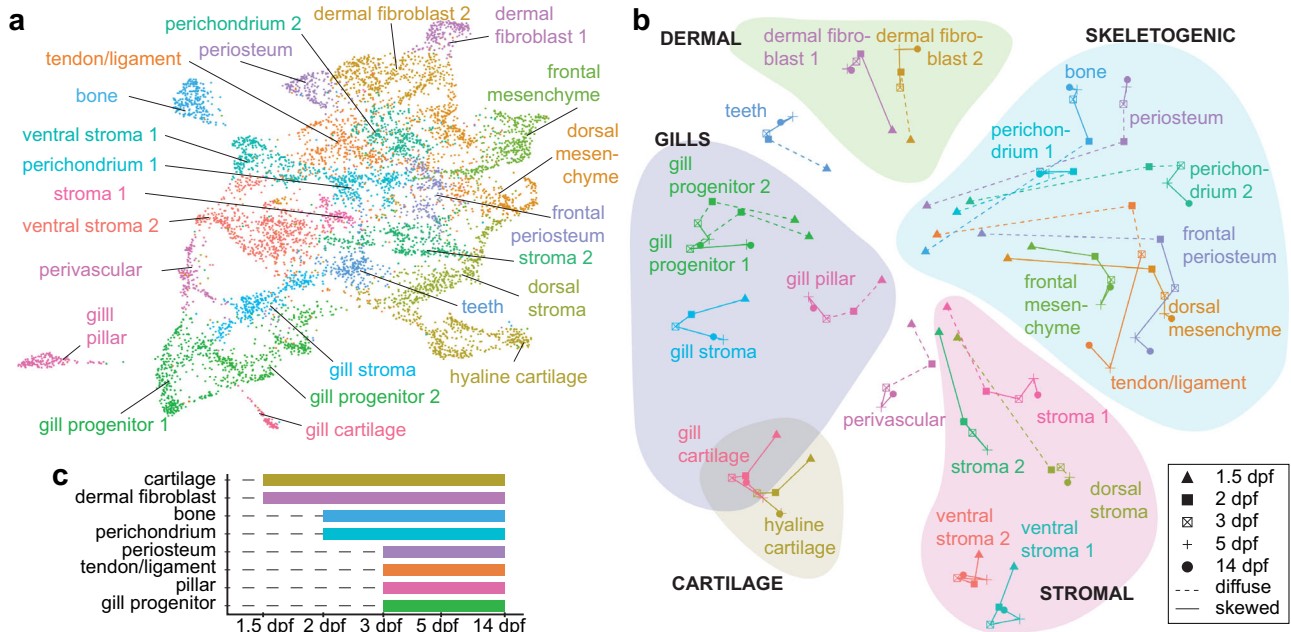

**Fig. 4 Cell type competency mapping through retrograde chromatin accessibility analysis. a** UMAP of 14 dpf SnapATAC data shows cell clusters for which top accessible peak modules were calculated for Constellations analysis. **b** Constellations analysis involves mapping of cluster-specific chromatin accessibility from 14 dpf back to earlier stages and then plotting relatedness of mapped accessibility in two dimensions. Diffuse refers to a stage when cluster-specific chromatin accessibility does not map to a discrete portion of UMAP space, and skewed when it does (see "Methods" for details). Groups of related cell types are color-coded. **c** Graphical representation from the Constellations analysis of when chromatin accessibility of major cell types first shows a skewed distribution in UMAP space, suggestive of establishment of competency.

marked by IRF8/*irf8*. Whereas loss of bone in mouse *Irf8*[−/−] mutants has been attributed to increased osteoclastogenesis[37], our analysis suggests that Irf8 may also have an early function in priming the skeletal lineage. Further, enrichment of CEBPA/ *cebpa* in stromal trajectories may reflect the immunomodulatory role of this mesenchymal population[29]. Of note, individual factors are not exclusively associated with particular cell types, and hence it is likely that particular combinations of factors drive lineage commitment (Supplementary Dataset 5). For example, ETS2 and GATA3 show strongest co-enrichment in the pillar cell trajectory, suggesting they could influence development of this unique type of gill-associated endothelial cell. These findings show how the Constellations analysis can reveal potential combinations of factors for establishing regional chromatin accessibility important for later cell type differentiation.

**Gill-specific lineages distinguished by early Gata3 activity**. Given the selective enrichment of GATA3 motifs and *gata3* activity in gill lineages, we further investigated the presence of a Gata3 regulatory circuit directing CNCCs to gill fates. Whereas previous work had shown that *gata3* is expressed in and required for initial gill bud formation in zebrafish, larval lethality had precluded analysis of gill subtype differentiation[38]. We find *gata3* expression to be maintained in gill populations through adult stages in scRNAseq data, which we validated by in situ hybridization at 14 dpf and 2 years of age (Fig. S23). We then identified a non-coding region ~143 kb downstream of the *gata3* gene, itself containing a predicted GATA3 binding site, that was selectively accessible in posterior arch CNCCs by 3 dpf, gill progenitors and pillar cells by 5 dpf, and gill cartilage cells by 14 dpf (Fig. 6a, Fig. S24). This *gata3-P1* element was sufficient to drive highly restricted GFP expression in posterior arch CNCCs starting at 1.5 dpf, which continued in gill progenitors, pillar cells, and chondrocytes through 60 dpf (Fig. 6c–e, Fig. S23).

Gill cartilage has a markedly distinct expression and chromatin accessibility profile from hyaline cartilage of the jaw, as shown by selective expression of *ucmaa* in gill cartilage versus *ucmab* in hyaline cartilage (Fig. S25). We identified a non-coding region ~5 kb upstream of the *ucmaa* gene that was selectively accessible in gill cartilage starting at 14 dpf and that contained a predicted GATA3-binding site (Fig. 6b, Fig. S24). This *ucmaa-P1* element drives highly restricted GFP expression in gill chondrocytes at 11 and 23 dpf, in contrast to a previously described *ucmab* enhancer[28] driving GFP expression in hyaline but not gill cartilage (Fig. 6f, Fig. S25). Although functional assays are needed to confirm Gata3 dependence, our findings are consistent with GATA factors establishing a positive autoregulatory circuit in posterior arch CNCCs that maintains *gata3* expression and promotes the later differentiation of gill-specific cell types (Fig. 6g).

## Discussion

Integration of transcriptome and chromatin accessibility data of the CNCC lineage has allowed us to connect patterning along major development axes to the emergence of the wide diversity of CNCC-derived cell types (Fig. 7). Rather than lineage-specific chromatin accessibility being established during CNCC specification, our Constellations analysis points to the progressive remodeling of chromatin underlying diverse cell type differentiation. Retrograde mapping of cell type-specific chromatin accessibility, combined with our highly resolved atlas of pharyngeal arch gene expression, also reveals candidate transcription factors for priming distinct CNCC lineages. Consistent with recent reports of organ-specific fibroblast heterogeneity[39], we uncover a distinct CNCC-derived dermal fibroblast population characterized by Phe/Tyr metabolism genes, which may be induced by early Wnt/Lef1 activity. Humans with mutations in *HGD*, which encodes an intermediate enzyme in the Phe/Tyr catabolic pathway, develop Alkaptonuria, or black bone disease,

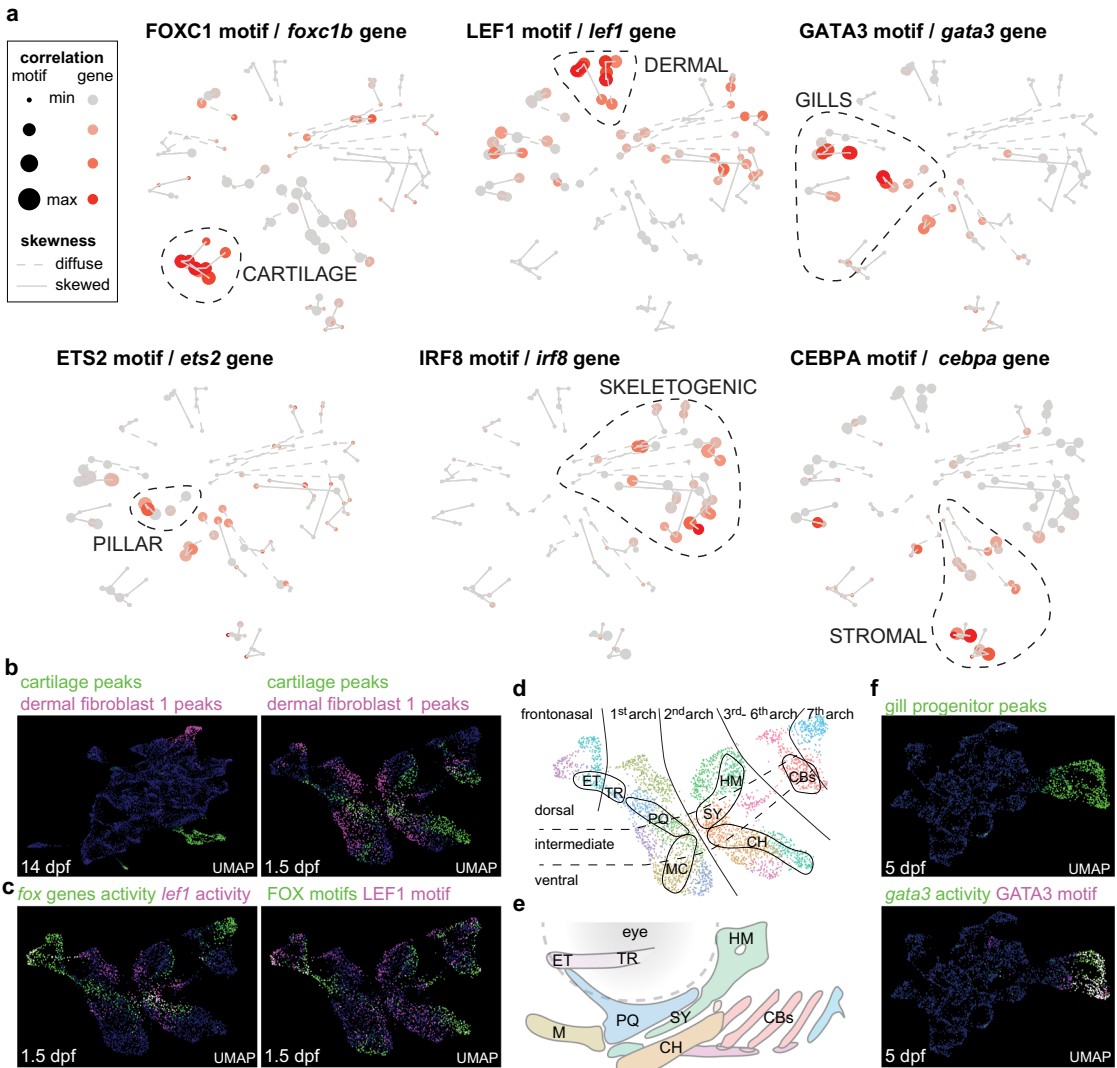

**Fig. 5 Constellations analysis reveals candidate transcription factors for lineage priming. a** Mapping onto the Constellations plot of transcription factors with correlated gene body activity and DNA-binding motif enrichment in specific clusters. Sizes of circles denote correlation of peak module mapping to motifs, and red color to gene body activities. **b** Top peak modules for hyaline cartilage and dermal fibroblast 1 clusters at 14 dpf mapped onto 14 dpf and 1.5 dpf SnapATAC UMAPs. **c** Summed gene body activities of *foxc1a*, *foxc1b*, *foxf1*, *foxf2a*, and *foxf2b* and summed FOXC1, FOXF1, and FOXF2 motifs at 1.5 dpf correlate with cartilage peak mapping, and *lef1* gene body activity and LEF1 motif with dermal fibroblast peak mapping. **d**, **e** Retrograde cartilage accessibility mapping at 1.5 dpf allows predictions of the arch origins of the individual cartilaginous elements of the week-old skeleton: ceratobranchials (CBs), ceratohyal (CH), ethmoid (ET), hyomandibula (HM), Meckel's (M), palatoquadrate (PQ), symplectic (SY), and trabecula (TR). **f** Mapping of the top peak module for 14 dpf gill progenitor clusters onto 5 dpf SnapATAC UMAP shows correlation with *gata3* gene body activity and GATA3 motif.

due to accumulation and pathological aggregation of homogentisic acid[16]. Expression of some of the same Phe/Tyr catabolic genes in a subset of axolotl limb fibroblasts[40] may point to broad and evolutionarily conserved roles for these specialized dermal fibroblasts in protecting the skeleton. We also identify largely distinct populations of *hyal4+* perichondrium cells, which represent putative skeletal progenitors in our pseudotime analysis, and *cxcl12a+* stromal cells. *Cxcl12+* stromal cells in murine bone marrow have been shown to only contribute to osteoblasts during bone regeneration[41]. It will therefore be interesting to test whether the *cxcl12a+* stromal population in animals such as zebrafish that lack bone marrow plays similar roles in skeletal regeneration[42]. In addition, we validate a *fgf10*-expressing progenitor population in the gill region that is characterized by sustained Gata3 activity, with later emergence of Ets2 activity in pillar cells providing a potential mechanism for the mesenchyme-to-endothelia transition of these specialized vascular cells. The

presence of a similar Fgf10-expressing mesenchyme population in the mammalian lung[43] raises the possibility that an ancestral CNCC-derived gill respiratory program may have been co-opted by the mesoderm during later lung evolution. Single-cell profiling of transcriptome and chromatin accessibility across time thus provides a blueprint for understanding the diversification and post-embryonic production of the huge variety of CNCC-derived cell types throughout the head.

## Methods

**Zebrafish lines**. The Institutional Animal Care and Use Committee of the University of Southern California approved all animal experiments (Protocol 20771). Experiments were performed on zebrafish (*Danio rerio*) of mixed AB/Tubingen background. For adult stages, the sex of animals is listed in Supplementary Dataset 1. Published lines include *Tg(Mmu.Sox10-Mmu.Fos:Cre)^zf384*[10]; *Tg(actab2:loxP-BFP-STOP-loxP-dsRed)^sd27*[44]; and *Tg(ucmab_p1:GFP)^el806*, *Tg(fli1a:eGFP)^y1*, and *Tg(sox10:DsRedExpress)^el10*[28]. Five transgenic lines were generated as part of this study: *Tg(fgf10b:nEOS)^el865*, *Tg(gata3_p1:GFP)^el857*, *Tg(gata3_p1:GFP)^el858*,

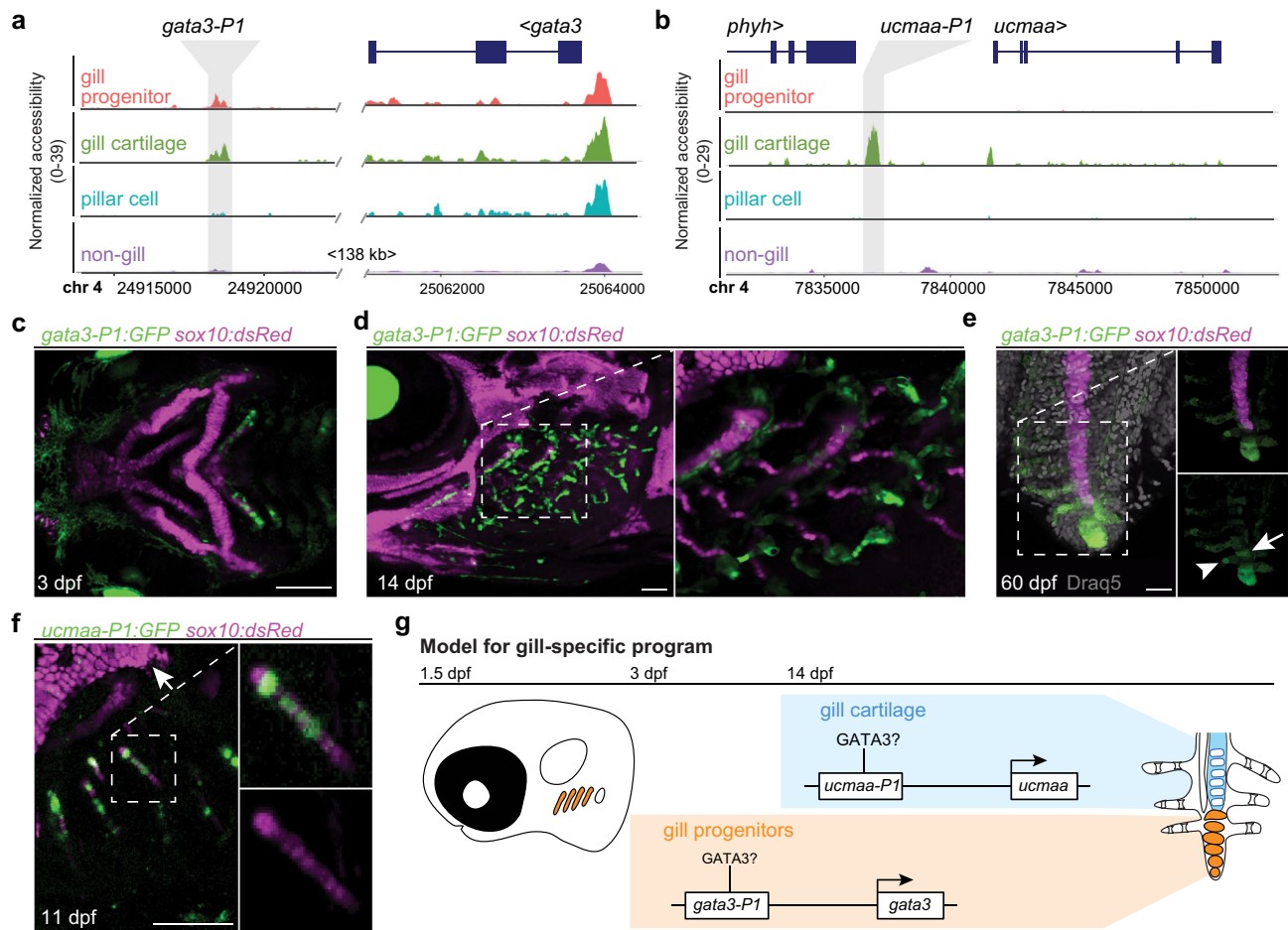

**Fig. 6 Gata3 activity distinguishes the gill-specific lineage. a, b** Genome tracks show chromatin accessibility for cell clusters from snATACseq data at 14 dpf. Gray shading shows gill-specific accessible regions near the *gata3* and *ucmaa* genes. Chromosome positions refer to the GRCz11 genome assembly. **c–f** *gata3-P1:GFP* drives expression in the posterior gill-forming arches at 3 dpf, the gill filament system at 14 dpf, and gill progenitors at the tips of primary filaments at 60 dpf, as well as some pillar cells (arrowhead) and gill chondrocytes (arrow) near the growing tips. *sox10:dsRed* labels cartilage and Draq5 nuclei. **f** *ucmaa-P1:GFP* drives highly restricted expression in *sox10:dsRed*+ gill chondrocytes (boxed region shown in merged and single channels to the right) but not hyaline cartilage (top left). **g** Model shows the initiation of *gata3* expression in the posterior gill-forming arches through the *gata3-P1* enhancer, maintenance of *gata3* in gill progenitors at the tips of growing filaments, and expression of *ucmaa* in gill cartilage through the *ucmaa-P1* enhancer. Both *gata3-P1* and *ucmaa-P1* contain predicted Gata3 binding sites. Scale bars, 100 μm (**c, d, f**), 20 μm (**e**).

*Tg(ucmaa_p1:GFP)el851*, and *Tg(ucmaa_p1:GFP)el854*. The *fgf10b:nEOS* knock-in line was made using CRISPR/Cas9-based integration[45]. Three gRNAs targeting sequences upstream of the *fgf10b* translational start site (5′-CATGA-TAACCCTTCCTAGAT-3′, 5′-GAGCTCTTTGATAGCGGGCT-3′, and 5′-GTTGAGCAGCATGTCCCATG-3′) were co-injected at 100 ng/μL into wild-type embryos with Cas9 RNA (100 ng/μL), an mbait-NLS-EOS plasmid (20 ng/μL)[46], and the published gRNA targeting the mbait sequence[45] to linearize the plasmid. A germline founder was identified based on nEOS fluorescence in the progeny of injected animals. For enhancer transgenic lines, we synthesized peaks for *gata3* (chr4:24918100–24918770) and *ucmaa* (chr4:7836670–783720) using IDT gBlocks and cloned these into a modified pDest2AB2 construct containing an E1b minimal promoter, GFP, and an eye-CFP selectable marker[28] using In-Fusion cloning (Takara Bio). We injected plasmids and Tol2 transposase RNA (30 ng/μL each) into one-cell stage zebrafish embryos, raised these animals, and screened for founders based on eye CFP expression in the progeny.

**In situ hybridization and immunohistochemistry.** All samples were prepared by fixation in 4% paraformaldehyde overnight, with 1.5 dpf embryos dehydrated in methanol for storage and animals for sectioning embedded in paraffin. Embedded animals were decalcified for one week in 20% EDTA if over 14 dpf. RNAscope probes were synthesized by Advanced Cell Diagnostics in channels 1 through 4. Channel 1 probes: *ifitm5, ucmaa, col10a1a*. Channel 2 probes: *postnb, myh11a, cxcl12a, sp7, gata3*. Channel 3 probes: *pah, lum, fgf10b, sox9a*. Channel 4 probes: *hyal4, acta2, ncam3*. Paraformaldehyde-fixed paraffin-embedded sections were deparaffinized, and the RNAscope Fluorescent Multiplex V2 Assay was performed according to manufacturer's protocols with the ACD HybEZ Hybridization oven.

For colorimetric in situ hybridization, paraffin sections were deparaffinized, blocked with Roche blocking powder (Roche, 11096176001), and digoxigenin-labeled riboprobes detected with anti-DIG-AP (1:1000, Roche, 11093274910). Visualization was with BM purple (Sigma, 11442074001). For whole-mount TSA fluorescent in situ hybridization at 1.5 dpf, embryos were treated with 1 μg/mL of proteinase K for 30 min (after acetone if combining with immunostaining). Probe hybridization was at 68 °C overnight. Digoxigenin-labeled riboprobes were detected with anti-DIG-HRP (1:500, Perkin Elmer, NEF832001EA) and visualized using Tyr-Fluorescein (1:100, Akoya, NEL701A001KT). The *hpdb* riboprobe was generated by cloning a purchased gBlock fragment designed from transcript hpdb-201 using nucleotides 679–1395 (tggatga…gactccc) into pCR-BluntII-TOPO (Thermo, K280040). The *nr5a2* riboprobe was generated by PCR amplification of cDNA with primers 5′-ATGGGGAACAGGGGCATATG-3′ and 5′-AGGGGTCGGGA-TACTCTGAT-3′, the *ucmaa* riboprobe with primers 5′-TGGTACCAGCTCAA-GACACT-3′ and 5′-ATAGTACTGGCGGTGGTGAG-3′, the *ucmab* riboprobe with primers 5′-ATGTCCTGGACTCAACCTGC-3′ and 5′-GTTATCTCC-CAGCGTGTCCA-3′, and the *thbs4a* riboprobe with primers 5′-CCCATGTTTCTTCGGTGTGA-3′ and 5′-GGTTTGGTACCAGCCTACAG-3′. Amplified products were cloned into pCR-BluntII-TOPO. pCR-BluntII-TOPO plasmids were linearized by restriction digestion (enzyme dependent on direction of blunt insertion), and RNA probes were synthesized using either T7 (Roche, 10881775001) or Sp6 polymerase (Roche, 11487671001) depending on direction of blunt insertion. Whole-mount immunohistochemistry for dsRed on 1.5 dpf embryos was performed with a 7 min −20 °C 100% acetone antigen retrieval and blocking in 2% normal goat serum (Jackson ImmunoResearch, cat. no. 005-000-121). Primary antibodies include rabbit anti-mCherry (1:200, Rockland Immuno-chemicals, cat. no. RL600-401-P16) and rabbit anti-mCherry (1:200, Novus

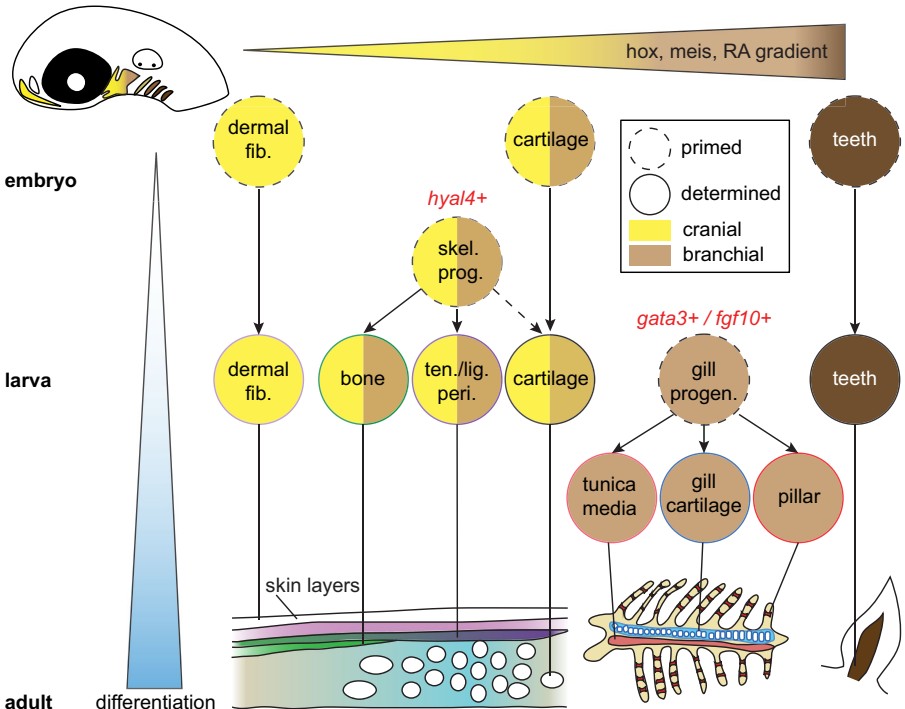

**Fig. 7 Model of ectomesenchyme lineages of CNCCs.** Cell types progressively emerge at distinct positions along the anterior-posterior axis of the developing head. Time is represented along the *y*-axis from young (white) to old (blue). Positional information is represented along the *x*-axis from anterior (yellow) to posterior (brown). Cell types are denoted by circles, with skeletal progenitors distinguished by *hyal4* expression and gill progenitors by *gata3* and *fgf10* gene expression. Mature cell types are schematized at the bottom.

Biologicals, cat. no. NBP2-25157) used at the same time to detect dsRed. The secondary antibody was goat anti-rabbit Alexa Fluor 568 (1:500, Thermo Fisher, A11035).

**Imaging**. Confocal images of whole-mount or section fluorescent in situ hybridizations and live transgenic fish were captured on a Zeiss LSM800 microscope using ZEN software. Colorimetric in situs were imaged on a Zeiss AxioScan Z.1. For *fgf10b:nEOS* experiments, we used the ROI function on the confocal microscope to specifically convert nEOS-expressing cells in the gill filaments of live animals using targeted UV irradiation, prior to the emergence of gill filament cartilage. At the specified days post-conversion, we euthanized the animal, fixed it in 4% PFA for 1 h, and dissected the gill arches. We stained the gills with DRAQ5 nuclear dye (Abcam) for 30 min and imaged at 40X to locate converted cells.

**Single-cell analysis**
*scRNAseq and snATACseq library preparation and alignment*. Heads from converted *Sox10:Cre; actab2:loxP-BFP-STOP-loxP-dsRed* fish were decapitated at the level of the pectoral fin at all stages (1.5–210 dpf). Eyes and brains were removed (eyes for 3–210 dpf, brains for 60–210 dpf). Dissected heads were then incubated in fresh Ringer's solution for 5–10 min, followed by mechanical and enzymatic dissociation by pipetting every 5 min in protease solution (0.25% trypsin (Life Technologies, 15090-046), 1 mM EDTA, and 400 mg/mL Collagenase D (Sigma, 11088882001) in PBS) and incubated at 28.5 °C for 20–30 min or until full dissociation. Reaction was stopped by adding 6× stop solution (6 mM $CaCl_2$ and 30% fetal bovine serum (FBS) in PBS). Cells were pelleted (376 × *g*, 5 min, 4 °C) and resuspended in suspension media (1% FBS, 0.8 mM $CaCl_2$, 50 U/mL penicillin, and 0.05 mg/mL streptomycin (Sigma-Aldrich, St. Louis, MO) in phenol red-free Leibovitz's L15 medium (Life Technologies)) twice. Final volumes of 500 μL resuspended cells were placed on ice and fluorescence-activated cell sorted (FACS) to isolate live cells that excluded the cytoplasmic stain Zombie green (BioLegend, 423111); see Supplementary Fig. 26 for gating strategy. For scRNAseq library construction, barcoded single-cell cDNA libraries were synthesized using 10X Genomics Chromium Single Cell 3′ Library and Gel Bead Kit v.2 per the manufacturer's instructions. Libraries were sequenced on Illumina NextSeq or HiSeq machines at a depth of at least 1,000,000 reads per cell for each library. Read2 was extended from 98 cycles, per the manufacture's instruction, to 126 cycles for higher coverage. Cellranger v3.0.0 (10X Genomics) was used for alignment against GRCz11 (built with GRCz11.fa and GRCz11.98.gtf) and gene-by-cell count matrices were generated with default parameters.

For snATACseq library construction, we used the same cell dissociation and sorting protocol as for scRNAseq, with live cells that excluded the cytoplasmic stain Zombie green (BioLegend, 423111) collected in 0.04% BSA/PBS. Nuclei isolation was performed per manufacturer's instructions (10X Genomics, protocol CG000169). Cells were incubated with lysis buffer on ice for 90 s, followed by integrity check of nuclei under a fluorescence microscope with DAPI before library synthesis. Barcoded single-nuclei ATAC libraries were synthesized using 10X Genomics Chromium Single Cell ATAC Reagent Kit v1.1 per manufacturer's instructions. Libraries were sequenced on Illumina NextSeq or HiSeq machines at a depth of at least 75,000 reads per nucleus for each library. Both read1 and read2 were extended from 50 cycles, per the manufacture's instruction, to 65 cycles for higher coverage. Cellranger ATAC v1.2.0 (10X Genomics) was used for alignment against GRCz11 (built with GRCz11.fa, JASPAR2020, and GRCz11.98.gtf), peak calling, and peak-by-cell count matrix generation with default parameters.

*SnapATAC for peak refinement and gene activity matrix imputation*. To refine the peak profile for better representation of diverse cell types across libraries, we performed a second round of peak calling using Snaptools (v1.2.7) and SnapATAC (v1.0.0) packages[17]. We first removed low-quality cells and cell doublets by setting cutoffs based on the percentage of reads in peaks (>30 for 60 dpf, >45 for 210 dpf, and >50 for the rest) and fragment number within peaks (5000–30,000 for 5 dpf, 1000–11,000 for 14 dpf, and 1000–20,000 for the rest). Potential cell debris or low-quality cells were removed by setting hard fragments-in-peak number cutoffs. Using the SnapATAC package, we then generated pseudo-multiome data at each stage. To recover every aligned fragment, we binned the genome into 5 kb sections and constructed the bin-by-cell matrices (bmats) for each library by Snaptools from the positional-sorted bam files generated by Cellranger ATAC v1.2.0. The cells were filtered, dimensionally reduced by diffusion map, and clustered with inputs of the first 34 dimensions following the SnapATAC vignette. The specific peaks were called for each cluster by the wrapped MACS2 function in SnapATAC with parameter gsize = 1.5e9, shift = 100, ext = 200, and qval = 5e−2. The finalized and refined peak profile was derived by collapsing and merging all 175 individual peak files to 445,307 peaks. To impute the gene activity with the corresponding scRNAseq data, the bmats of each time point were used to calculate gene-activity-by-cell matrices (gmats) by SnapATAC. The gmats were used to find anchors within the scRNAseq data at the same time point by Seurat. We then transferred the expression data from scRNAseq through the anchors to derive the imputed gene-activity-by-cell matrices for each time point.

*Data processing of scRNAseq and snATACseq*. The count matrices of both scRNAseq and snATACseq data were analyzed by R package Seurat (v3.2.3) and Signac (v1.0.0). The count matrices of each sample were aggregated where replicates were available. For scRNAseq data, the matrices were normalized (NormalizeData) and scaled for the top 2000 variable genes (FindVariableFeatures and

ScaleData). The scaled matrices were dimensionally reduced to 30 principal components (60 components for 150 dpf) using Seurat functions JackStrawPlot and ElbowPlot, in addition to prior biological knowledge of the craniofacial structures. The data were then subjected to neighbor finding (FindNeighbors, k = 20) and clustering (FindClusters, resolution = 0.8). The data were visualized through UMAP with 30 principal components as input. For snATACseq data, the matrices were dimensionally reduced to 30 latent semantic indices (LSIs) through RunT-FIDF and RunSVD functions. The neighbor finding, clustering, and visualization were performed as for scRNAseq data (algorithm = 3 for FindClusters with resolution = 0.8, and resolution = 1 for 14 dpf) with input of the second to thirtieth LSIs. To confirm the dataset was not over-clustered under these resolutions, we performed subsampling-based approach chooseR to test robustness[47]. Enrichment of motifs included in JASPAR2020[48] in accessible chromatin regions was calculated by chromVAR[49] using function RunChromVAR. To test the enriched genes and gene activities in both scRNAseq and snATACseq data, two-sided likelihood-ratio tests were performed through the FindAllMarkers function (min.pct = 0.25) with cutoff of adjusted $p$ value smaller than 0.001.

*STITCH network construction and force directed layout.* To identify the overall cell trajectories in our scRNAseq and snATACseq data, we used the STITCH algorithm[18] to construct cell neighbor networks in MATLAB (R2018b). As the dimensional reduction space of snATACseq data are LSIs, we modified the stitch_get_knn and stitch_get_link function of the STITCH package to make it compatible with LSI. For stitch_get_knn function of snATACseq data, we used the LSI matrix to find the k nearest neighbor of each cell for each time point. For stitch_get_link function of snATACseq data, we projected the LSI space of time point $t$ to time point $t-1$ by solving the right orthogonal matrix of the singular vector decomposition (SVD) for $t-1$. SVD is the initial step of latent semantic analysis where $M$ is the peak-by-cell matrix and $U$ will be later transformed to LSI. For $t$ and $t-1$, we have

$$M_t = U_t \Sigma_t V_t^T \quad (1)$$

$$M_{t-1} = U_{t-1} \Sigma_{t-1} V_{t-1}^T \quad (2)$$

To project the space from $t$ to $t-1$, we derived a projected $U_{t-1}$ as $U_{t-1}^p$ by solving the equation.

$$M_{t-1} = U_{t-1}^p \Sigma_t V_t^T \quad (3)$$

Both $U_t$ and $U_{t-1}^p$ were further combined, normalized, and subjected to the default neighbor finding as stitch_get_link. To visualize the STITCH networks of both scRNAseq and snATACseq data, we used the force directed layout by ForceAtlas2 in Gephi (v0.9.2) to derive the two-dimensional layouts.

*Pseudotime analysis.* We used the R package monocle3 (v0.2.3.0) to predict the pseudotemporal relationships within jaw or gill subsets. We first merged 5 and 14 dpf scRNAseq data, including an additional scRNAseq library of *Sox10>dsRed+* cells sorted from the dissected ceratohyal endochondral bone at 14 dpf (to further enrich skeletogenic populations), and performed clustering and dimensionality reduction (60 principal components, k = 25, resolution = 1). After removing non-mesenchyme populations, and dermal fibroblast (*pah+*) and tooth mesenchyme (*spock3+*) clusters, we placed *hoxb3a+/gata3+* cells into a gill cluster and all other cells into a jaw cluster. Cell paths were predicted by the learn_graph function of monocle3. We set the origin of the cell paths based on the enriched distribution of 5 dpf cells.

*Gene ontology, motif family, and transcription factor (TF) analysis of CNCC mesenchyme.* Analysis was performed on ectomesenchyme, perichondrium, gill progenitor, and stromal populations at each stage based on markers from the scRNAseq data (Figs. S1–7). The enriched genes of each cluster are tested by running a two-sided Wilcoxon rank sum test against all other clusters with cutoff of adjusted $p$ value ≤ 0.001. These enriched gene sets are subjected to gene ontology analysis for terms of biological processes (BP) by R package ViSEAGO (v1.2.0)[50]. The heatmap is generated by GOterms_heatmap function using values of log10(adjusted $p$ value). To generate the heatmap of motif families for each cluster, we first averaged and aggregated the motif accessibilities for each cell according to the motif family by TRANSFAC[51]. The means of each motif family are used for the heatmap. To generate the heatmap of TFs and chromatin remodeling factors for each cluster, we subsetted the TFs and chromatin remodeling factors from the enriched gene sets, and we used the mean of each TF and chromatin remodeling factor for every cluster for the heatmap.

*Constellations analysis and calculation of cluster skewedness and correlation.* The tissue module scores of the snATACseq data were calculated based on the enriched peak sets and their module scores for each cluster identified at 14 dpf by R packages Seurat and Signac. The enriched peak sets were calculated by the FindAllMarkers function using two-sided likelihood ratio tests with fragment numbers in peak region as latent variables. We used the peaks with adjusted $p$ values smaller than 0.001 as the enriched peaks for a cluster. As there are 23 clusters (tissues) identified

at 14 dpf, we ended up with 23 peak sets, which we applied to earlier time points (1.5, 2, 3, and 5 dpf) to calculate the tissue module scores using AddChromatinModule function. To determine whether a tissue score at a time point distributes in a statistically significant, and hence biologically interesting, way, we calculated the skewedness of the distribution of a tissue score by the R package parameter (v0.12.0). We considered a tissue score to be distributed in a meaningful way if it was strongly right skewed by a hard cutoff of skewedness greater than 1. For 1.5 dpf, the cutoff of skewedness was lowered to 0.4 to accommodate overall lower skewedness at that time point, but with additional filter of max module score >15 to avoid tissue module scores with extremely low values.

To profile the relationship of all tissue scores, we constructed a distance matrix of all 23 tissue module scores across all the time points (1.5, 2, 3, 5, and 14 dpf). For the distance $D$ between score of tissue $A$ at time point $t_1$ and score of tissue $B$ at time point $t_2$, the distance $D$ can be described as

$$D = D_{\text{tissue}} + a \times D_{\text{timepoint}} \quad (4)$$

$$D_{\text{tissue}} = [d_{(A,B),t_1} + d_{(A,B),t_2}]/2 \quad (5)$$

Where $d_{(A,B),t_1}$ and $d_{(A,B),t_2}$ stand for the Euclidean distance between score $A$ and $B$ at time point $t_1$ and $t_2$, respectively.

$$D_{\text{timepoint}} = d_{(t_1,t_2)} \quad (6)$$

Where $d_{(t_1,t_2)}$ stands for the Robinson-Foulds distance between the tissue score dendrogram of time point $t_1$ and $t_2$. Since $D_{\text{timepoint}}$ is relatively smaller than $D_{\text{tissue}}$, we multiply $D_{\text{timepoint}}$ by 12 ($a = 12$) to make the distance between time points comparable to the distance between tissue scores. The distance matrix was dimensionally reduced and visualized by UMAP.

To detect the potential factors that contribute to the patterning of tissue-specific peaks, we performed linear regression of each tissue module score against all motif accessibilities and the gene activities of transcription factors. We used ZFIN and JASPAR2020, converted by homology data from MGI, to build up a list of transcription factors in zebrafish. We then curated and paired every motif in JASPAR2020 with its potential binding transcription factors. The coefficients of regression results were used as indications of whether a motif or transcription factor is positively correlated with a tissue module score with upper cutoff of adjusted $p$ value 0.05. We transformed the coefficients of all the negative related motifs and transcription factors to 0 to filter out irrelevant motifs and transcription factors. To visualize the correlation of each pair of motif and transcription factor, we plotted the coefficient magnitudes of motifs by dot sizes and transcription factor gene body activities by a red color scale on Constellation maps.

**Statistics and reproducibility.** We included biological replicates at several stages to test the reproducibility of library preparation and increase depth of data. For scRNAseq, we performed two replicates at 5 and 14 dpf, and three replicates at 3 and 150 dpf. For snATACseq, we performed two replicates at 2, 3, and 14 dpf. All in situ patterns were confirmed in at least 3 independent animals. Two independent germline founders were identified for each transgenic line and in each case were verified to show similarly specific activity in the gills. For all transgenic imaging experiments, expression patterns were confirmed in at least 5 independent animals. For juvenile and adult lineage tracing experiments, results were confirmed in at least 2 independent experiments performed on different days. No statistical method was used to predetermine sample size. No data were excluded from the analyses, and the experiments were not randomized. The investigators were not blinded to allocation during experiments and outcome assessment.

**Reporting summary.** Further information on research design is available in the Nature Research Reporting Summary linked to this article.

## Data availability
The scRNAseq and snATACseq datasets generated in this study have been deposited in the Gene Expression Omnibus (GEO) database under accession code "GSE178969". The processed Signac/Seurat rds objects for the scRNAseq and snATACseq data are available at FaceBase under record key "5-DAQ4". The STITCH file connecting all scRNAseq datasets is available "on SPRING viewer [https://kleintools.hms.harvard.edu/tools/springViewer_1_6_dev.html?client_datasets/Peter_V3_080918_nbrm/Peter_V3_080918_nbrm]". All other relevant data supporting the key findings of this study are available within the article and its Supplementary Information files or from the corresponding author upon reasonable request.

## Code availability
Code for recapitulating the results in this manuscript have been released at "Github [https://github.com/TedKCT/Crump_zebrafish_scNCC/releases/tag/v1.0]"[52].

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

## Acknowledgements

We thank Megan Matsutani and Jennifer DeKoeyer Crump for fish care, Jeffrey Boyd and the BCC FACS core, the CHLA Sequencing core, the USC Center for Advanced Research Computing, and Andrew McMahon, Yang Chai, and Unmesh Jadhav for helpful comments on the manuscript. Funding was provided by the following grants from National Institute of Dental and Craniofacial Research of NIH: R35 DE027550 (J.G.C.), K99 DE029858 (P.F.), F31 DE029682 (C.A.), F31 DE030706 (M.T.), and R00 DE027218 (J.S.), as well as NICHD T32 HD060549 (M.T.).

## Author contributions

P.F., K.-C.T., M.T., C.A., H.-J.C., J.S., N.N., and J.G.C. performed the experiments. J.G.C. oversaw the project and wrote the manuscript. K.-C.T. generated code for the Constellations analysis. P.F., K.-C.T., M.T., C.A., and J.G.C. interpreted the data.

## Competing interests

The authors declare no competing interests.
