## [Peer Review File · Nature Communications]

Reviewers' Comments:

Reviewer #1:

Remarks to the Author:

In this paper, the authors have generated several large and robust datasets detailing the differentiation trajectories of cranial neural crest cells (CNCC) over a wide gulf of developmental stages, spanning from embryonic to adult at the single cell level in zebrafish. They generated both RNA sequencing datasets as well as ATACseq datasets, which they leveraged to discern both active cell identity and their coordinated regions of chromatin remodeling as cranial neural crest make cell fate decisions. The authors refreshingly took time to clearly demonstrate that their datasets are congruent with the expected results obtained from other well characterized transgenic lines (fli1a:eGFP and sox10:dsRed). They identified an early divergence around 3 dpf between skeletal and gill mesenchyme, of interest to the authors, as well as several additional lineages which received less focus including teeth, otic placodes, and dermal fibroblasts. The authors present an exciting novel transgenic zebrafish line Tg(fgf10b:nEOS) and demonstrated novel evidence that fgf10b+ cells form specialized gill stem populations, which becomes increasingly restricted to the distal tips of the gill mesenchyme with age. This population differed from that of the hyal4+ populations which gave rise to skeletogenic lineages. For both of these major lineages, large swaths of genetic motifs and specific transcriptional trends were identified, though the authors missed a valuable opportunity to highlight in these remarkable analysis novel relationships between the various lineages they traced.

The authors leverage a pre-existing pipeline called SnapATAC to merge multistage snATACseq datasets with annotated scRNAseq datasets. They successfully identify several genes whose expression coordinately changes through time along with the accessibility of their chromatin regions. Through this method and motif analysis, they note several transcription factor families which were active during the differentiation of CNCC lineages. Notably, they were able to confirm the presence of two previously annotated false-positive genes, highlighting the power and sensitivity of their approach. The merged dataset is passed through to a novel "Constellation" pipeline. Summarily, the authors found coordinated sequences at open chromatin regions in the most differentiated datasets and passed these signatures to early clusters of cells, calculating their relatedness in two-dimensional space. These formed reduced constellations plots, reminiscent of PAGA plots (Wolf et al., 2019), to represent relatedness of clusters. It is not clear why the authors did not simply use PAGA for their analysis, but instead contrived a seemingly less robust approach. As noted below, the input clusters used to specify the cluster-specific peak modules which define the crux of this analysis were not robustly tested. Despite this issue, using these Constellation plots several core motifs were identified pertaining to five major tissue lineages, which many of which were confirmatory of the literature. In an interesting finding, the authors discern that GATA3 persistently defines the presumptive gill cartilage progenitors through the use of a novel transgenic line. Cumulatively, the authors posit that these data support a model in which CNCC are primed for fate acquisition through the regulation of chromatin accessibility in a selectively restrictive fashion during differentiation in multiple discrete lineages.

In conclusion, this paper encompasses an extensive body of work which is of extraordinarily high significance to the field. Despite the very limited shortcomings, this paper is deserving of very high praise. While not quite publishable in its current form, after the issues are addressed, this work will greatly advance the field and serve as a touchstone for future analysis of cranial neural crest.

Major Issues:

1. The authors have utilized the STITCH to conduct their merged analysis in some respects though in other places they interchangeably use UMAPS without rationale provided for the switch. Justification should be provided for their usage. While UMAPS preserve some ability to intuitively discern the relationship between clusters based on their distance in 2D, it is unclear from both the current documentation in the STITCH package and the original publication if the STITCH plot retains this feature.
2. In the generation of the scRNAseq datasets, justification should be provided as to why the number of principle components was used for the determination of scaling and subsequent analysis.
3. The technical advantage of the authors Constellation analysis over the widely used and well established PAGA plot method is not abundantly clear from the text. At the surface level, this technique appears to be underpowered compared to PAGA, which does not require the iterative

computation of a two-sided likelihood tests, decreasing the false positive rate. The authors should address why PAGA is insufficient or inadaptatable to perform their analysis.

4. Regarding lines 87-90, a supplementary table should be prepared which describes in detail the fractions of cells which were collected from the FACS, the number of cells loaded per 10x analysis, the number of cells retained after thresholding per library, the median genes per cell per library, and the actual read depth per library. Similar relevant data should be provided to support the snATACseq datasets. These technical statistics will build confidence in the generation of the author's datasets as well as ensure the reproducibility and interpretability of this study in the future. Additionally, the number of animals used per time point collected per replicate, as well as information as to the sex of older animals should be reported, possibly to also be included in this table.

5. Surprisingly, there is no discussion or search for chromatin remodeling factors anywhere in the results. This analysis would greatly enhance the strength of the conclusions and should be included in some form. If not, if the authors could explain why, that would be helpful.

6. At line 94, A table should be included clearly listing the specific marker genes utilized to categorize each cluster in each dataset as well as appropriate citations to support each gene's use. Current reporting of total genes listed per cluster in table 1 is valuable and should not be removed.

7. At line 392 (referenced at line 83-84), the specific boundaries utilized to define the cranial domain collected per time point should be included.

8. The authors assume that the number of clusters demonstrated in their 14 dpf dataset, which underpins much of their analysis, is representative of 23 distinct tissue classes as they identified 23 clusters. However, the number of clusters will robustly shift depending on the resolution parameter set during the analysis. This assumption may impact the degree of sensitivity of this analysis, by which many conclusions in the paper are drawn. Rationale is required for the arbitrary setting of the 0.8 resolution parameter by which the clustering was formed. Further, the robustness of their novel constellation analysis should be demonstrated at a range of resolution parameters. Alternatively, if the authors have data demonstrating definitively that 23 separate tissue lineages are present at 14 dpf would also support their case.

9. At line 235, regarding the establishment of the constellation analysis, it is not clear why the authors cut off their analysis at 14 dpf when additional datasets at 60 dpf and 150 dpf were also at hand. Validation of the results from the constellation pipeline including these datasets would be a dramatic improvement to the current version of the analysis.

10. In somewhat of a continuation of the prior comment, at line 105, the authors claim to find "17 distinct clusters" following their analysis in Seurat. However, no discussion has been offered about how clusters were determined to be accurate, undermining this claim. An explanation as to how the authors arrived at the appropriate number of clusters using the 0.8 resolution parameter is needed.

11. The conclusions drawn from the motif enrichment analysis discussed in lines 268-278 is a very light treatment, failing to reflect much of the nuance of expression of the transcription factors in multiple lineages. For example, the ETS2 motif is only addressed in the context of the pillar cells, while it clearly is also expressed in the stromal constellation as well. Additionally LEF1 is characterized within the Dermal constellation when there is involvement in the skeletogenic and gill constellations. In total, the authors have truly uncovered a combinatorial code of these transcription factors paired with specific open chromatin motifs. This astonishing achievement would be better highlighted in this text, rather than relying on such a heuristic approach to the interpretation of these graphs.

12. I am confused how approximately 50% of the merged scRNAseq atlas that is present in Figure 2a can be unidentified when there are assigned lineage identities to every population that is present in the individual datasets presented in the supporting supplemental figures. Further, labels applied in the supplements do not correspond to the labels used to describe lineages in the atlas. The authors should endeavor to make these labels congruent to help facilitate the reader's ability to see the source for each cell lineages shown in the merged STITCH plot.

13. Paragraph from line 216-230 is a superb treatment of characterizing the findings of the SnapATAC data within the context of the literature, but fails to leverage the true power of this dataset to identify novel or unexpected motifs. Please include some analysis or discussion on this matter.

14. Beginning at line 517, the use definition of equations is confusing and circuitous. Standardized equation should be used when possible. Further, at line 520, the authors define $D(\text{tissue})$ with text rather than through a standardized equation format. A similar issue occurs at line 522. Lastly, I

- believe the authors intended to define a as 8 at line 525, though it is unclear. Care should be taken to revise this paragraph for clarity as it is essential to their Constellation method.
15. At line 129, the Authors claim that their evidence is clearly apparent for snATACseq, but then explain all their findings using the scRNAseq plots without clarifying this point. Please address this discrepancy or change the claim to reflect the scRNAseq.
 16. The authors claim at line 132 that the *fgf10b*+ population is present at 5 dpf and direct the reader to Figure S10, but do not show data relevant to this time point. Please include micrographs reflecting these findings in 5 dpf animals.
 17. Same comment above, regarding data discussed in lines 159-162, please show 5 dpf/14 dpf animals.
 18. Line 134-135 the claim that no cardiomyocytes is found is very misleading as the population of NCC sampled should not include these cells. Further, the supporting figure does not support this claim. Please strike this clause.
 19. At line 140, it is difficult to interpret the lack of expression within datasets not fully knowing the read depth of each dataset and given the relatively low genes/cell value reported. How are the authors sure that this inconsistency in their dataset with prior literature, which they cite, not point to a more systemic issue with the resolution of their datasets?
 20. Line 152, the authors claim that there are two distinct lineages, one of which is defined by *cxcl12a*+/*ccl25b*+ and the other by solely by *hyal4*+ cells, however there is clearly a *hyal4*+/*cxcl12a*+ domain shown in figure 2e. Please address this discrepancy.

Minor Issues:

1. At line 67, the authors state that "These [prior] findings point to extrinsic inductive cues of CNCC fate determination," which does not seem to be in line with the hypothesis they propose on line 68. Perhaps a clause is missing to connect the two thoughts?
2. The authors do a remarkable job presenting a vast amount of data in a short span. Summary statements at the end of each major section would make this dense body of work more accessible.
3. In Figure 2i, the arrows do not point to cells that are new EOS+ cells, but rather to cells which were previously converted and have since continued to express *fgf10b*. On line 170, the authors state that the cells at the new pillar cells should be "green only," however cells shown in Figure 2i, at day 10 are clearly green and magenta. Either the text needs to be corrected to reflect the anticipated co-labeling of the cells at the tips or a new yellow arrow needs to be drawn to point to the intended population of cells.
4. At line 108-109, there is a redundant usage of "and" in the list of lineage derivatives.
5. In supplemental figures in which a single letter is used to designate a group of images, the inclusion of some sort of boundary (eg. a thin boxed outline or shaded background) to designate which panels specifically belong to that letter designation would greatly improve the figure clarity.
6. At line 110-112, the authors claim that this finding is "unexpected," but do not offer an explanation as to why. Please include a clause to make this point clear.
7. At line 116, citations are needed to support the use of these marker genes. May also be resolved through the inclusion of the genes in the marker gene table requested in the major issues section.
8. At line 148-149, the authors use specific gene combinations to define posterior arch mesenchyme but defer generally to a "skeletogenic" signature for their anterior arch mesenchyme. Please define the set of genes used to identify "skeletogenic" cells.
9. Equilateral triangles used as arrow heads obscure the direction intended to be seen in some micrographs. Change in arrowhead shape or the additional of tails will clarify meaning. Different arrow shapes are also used throughout the paper, which detracts from the style.
10. At line 152, in figure 2C-E, there is a strange aberrance in the way the pseudotime graph was learned. It appears that the presumptively more differentiated cell population at 14 dpf in the lower half of the gill stroma (Fig. S13e, red population) is leading to a 5 dpf population which is flagged as earlier according to the color scale. Please verify the correct branch node was used to establish to graph learning for Figure 2d.
11. Line 162, I am not convinced by these data alone that the *cxcl12a*+/*ccl25b*+ cells are not also contributing to the perivascular population, which is also proximal to the basal root of the pseudotime tree. Can you provide further support to address this possibility?
12. In Supplementary Figure 10d, the white text is nearly impossible to read. Please change color of text and micrograph label to improve visibility.

13. Line 167, please replace "over time" with a specific interval of time you assayed.
14. Around Line 465, please identify the programming environment used for this analysis as well as the version number if different from R.
15. Please clearly label UMAPS on black backgrounds to distinguish them easily from microscopy images.
16. In supplementary figure 11, 12, and 13, cell type labels above the STITCH graphs along with the gene name shown would greatly enhance the accessibility of these data, similar to the format used in Figure supplement 10.
17. Several paragraphs begin with the same formulaic introduction "To XXX, we YYY.," which undermined the exciting data contained within. Improvements in the writing style would make the flow of the paper more pleasant to read.
18. Starting at line 201 to the end of that paragraph, this speculation, while intriguing, should be placed in the discussion as it is not a relevant result.
19. Several intriguing speculations are presented throughout the study. Due to the discovery of phenylalanine and tyrosine catabolic pathways in CNCC-derived fibroblasts, the authors posit that these pathways may drive the etiology of Alkaptonuria. Further, the authors speculate that *cxcl12a+* populations in zebrafish contribute to skeletal regeneration, unlike in mouse models. Both of these speculations would be better suited for in the discussion rather than the results.
20. A commitment should be made in the Data Availability section at line 553 to provide access to the code used in the generation of this project.
21. In Supplement figure 14, the number animals differs from the number of animals reported in line 388. Please edit to reflect.
22. At line 406, please provide justification as to why 126 cycles was selected for this sequencing while only 65 cycles was selected at line 417 for the ATAC sequencing.
23. At line 453, if a work flow was utilized in the Seurat package to integrate the datasets, please explain which workflow was used.
24. Please include information about code generation and data interpretation in the author statement.

Wolf, F.A., Hamey, F.K., Plass, M., Solana, J., Dahlin, J.S., Göttgens, B., Rajewsky, N., Simon, L., and Theis, F.J. (2019). PAGA: graph abstraction reconciles clustering with trajectory inference through a topology preserving map of single cells. *2Genome Biol.* 20, 59.

Reviewer #2:

Remarks to the Author:

The authors have addressed the topic of how cranial neural crest cells (CNCCs) acquire their diverse final fates by generating a single cell transcriptome and chromatin (ATACseq) 'atlas' of cell fates across developmental time. The extended temporal sequence, through embryonic, larval and on into juvenile and adult stages of the zebrafish, is a key aspect of this study that extends it well beyond prior work in this and other models. Several interesting findings have emerged, such as the existence of a specialized type of gill cartilage, distinct from that elsewhere in the head, and a set of fibroblasts enriched in enzymes for phe/tyr breakdown. In situ hybridization/RNAscope has been used to validate some of the key expression patterns and the data are of high quality.

By establishing developmental trajectories, the authors find a progressive emergence of fates as early as 3dpf. The analysis points to progressive emergence of *hya1h+* perichondrium, *cxcl12a+* stromal, and *fgf10a/b+* gill populations. The authors also nicely confirm that *Fgf10b+* cells are progenitors for specialized gill cell types, by integrating a convertible label into the *fgf10b* locus and following the cells over time. This is both a convincing results and a useful proof or principle for similar analysis of other cell types.

The use of *Sox10*CRE +ve cells ensures that all the cells analyzed are NCCs (with the exception of a few otic vesicle cells where there is also expression). This makes for a clean study, but it leaves me confused about the overarching conclusion of the importance of extrinsic factors. Specifically, I find that while the data are compelling and well presented, the intellectual framework —both the rationale for the study and the overarching conclusions lacks equivalent clarity.

The authors essentially set up a dichotomy between intrinsic and extrinsic patterning at the onset of the paper, but do not lay out what experimental outcomes would support or refute each of these models. Moreover, whether intrinsic patterning is equivalent to the concept of "pre-patterning" is not discussed, and I would certainly appreciate citation of the chick transplant work of Drew Noden (1983) that led to the general idea of pre-patterning in the CNC cell population. Importantly, the term intrinsic can be used at either a single cell level or at a tissue level, and the authors do not make their meaning clear. However, as their approach focuses exclusively (and appropriately) on the CNCC lineage it obviously has less to say about the role of external tissues. Nevertheless, in either case, 'extrinsic' implies a role for adjacent cells and the secreted – or possibly membrane bound – molecules from those adjacent cell types that influence the cell type of interest. Thus, an important if indirect approach to recognizing extrinsic activity is a requirement for specific receptor function. While Wnt5 and Fgfa10 signals are touched upon, this overall idea is not well discussed.

Adding to the confusion are brief allusions to the work of the Gillis lab demonstrating that mesoderm-derived cells contribute to the skate gill skeleton. In the Discussion the author's comments appear to be hinting that these mesoderm-derived cells may come under a common influence together with adjacent CNCCs. If this is the authors' meaning then it needs to be spelled out, and note made that this is speculative rather than based on data. In the Introduction the same work from Gillis is cited as part of the rationale for the study, with the authors stating that these findings "point to extrinsic inductive cues for CNCC fate determination". At least as currently written I find this logic opaque.

Overall, I find this study to be well done, and an important addition to our knowledge, but I feel the writing needs some work, especially to explicate the underlying logic for the conclusions.

Reviewer #3:

Remarks to the Author:

This project conducts sc-seq and sc-ATAC on single cells derived from the neural crest in zebrafish at seven stages starting in the embryo and ending in the adult. These datasets, and the fusion of them, permits the identification of a large number of cell types including precursor cells at various stages of differentiation and differentiated cells themselves. New bioinformatic methods are applied (SNAP-ATAC, for fusing sc-seq and sc-ATAC, and STITCH for predicting developmental trajectories through time) and a new one is developed (CONSTELLATION for mapping cell cluster-specific chromatin accessibility).

Strengths of the paper include the fact that stages from embryo to adult are samples as part of a single study, which is the first of its kind. The use of both RNA seq and ATAC seq is a strength as together the data yield more accurate clusters. Finally, most single cell sequencing papers, even those in top tier journals, are essentially descriptive. They generate testable hypotheses but do not test many or any of them. The primary value of this paper, like the others, is that it provides a starting point for more detailed explorations of the neural crest derivatives and trajectories that gives rise to them. Particularly appealing is the retrograde mapping of transcription factor binding sites that emerge progressively along trajectories to particular derivatives. This information can be used to deduce a presumed hierarchies of the gene regulatory networks leading to each derivative. However, satisfyingly, it goes a step further conducts a lineage experiment to test a specific lineage relationship suggested by the sequencing data. Those data suggest that fgf10+ cells are the origin of 4 specialized gill subtypes. They generate fgf10+:eos line and use it follow the fate of cells that expressed fgf10+. They discover that the lineage label ends up in two of the 4 subtypes (gill cartilage and pillar cells). What about the other two gill subtypes – tunica media and perivascular populations? Some comment is merited here.

I felt that the manuscript suffered a bit from the style of the writing. Stronger topic sentences that emphasized the main conclusion presented in each paragraph would make the central points of the study easier to understand. When I first read it I felt that I had learned a little bit about a lot of things.

REVIEWER #1:

- This paper encompasses an extensive body of work which is of extraordinarily high significance to the field. Despite the very limited shortcomings, this paper is deserving of very high praise. While not quite publishable in its current form, after the issues are addressed, this work will greatly advance the field and serve as a touchstone for future analysis of cranial neural crest.

Response: We thank the reviewer for these kind words.

1.1 - The authors have utilized the STITCH to conduct their merged analysis in some respects though in other places they interchangeably use UMAPs without rationale provided for the switch. Justification should be provided for their usage. While UMAPs preserve some ability to intuitively discern the relationship between clusters based on their distance in 2D, it is unclear from both the current documentation in the STITCH package and the original publication if the STITCH plot retains this feature.

Response: Both UMAP and force-directed layout, used by STITCH, preserve some ability to discern the relatedness of cell clusters on a 2D layout. UMAP performs better at distinguishing discrete clusters at each of the stages analyzed. In contrast, the force-directed layout of STITCH highlights distinct cell trajectories over time. We therefore use UMAPs for single time-points and STITCH when visualizing trajectories over the entire time series. We have modified the text to clarify that STITCH allows visualization of both cell relationships and trajectories on a 2D layout.

Lines 132-134: "First, we applied the STITCH algorithm¹⁸ to scRNAseq and snATACseq datasets to connect individual stages; the force-directed layout of STITCH allows visualization of both cell relatedness and developmental trajectories in 2D space (Fig. 2a,b)."

1.2 - In the generation of the scRNAseq datasets, justification should be provided as to why the number of principle components was used for the determination of scaling and subsequent analysis.

Response: We now explain in the Methods that we used JackStrawPlot() and ElbowPlot() in Seurat and prior biological knowledge of the craniofacial structures, to determine the number of principle components. We also confirmed the dataset was not over-clustered under these resolutions by performing subsampling-based approach "chooseR" as described in Patterson-Cross et al., 2021.

1.3 - The technical advantage of the authors Constellation analysis over the widely used and well established PAGA plot method is not abundantly clear from the text. At the surface level, this technique appears to be underpowered compared to PAGA, which does not require the iterative computation of a two-sided likelihood tests, decreasing the false positive rate. The authors should address why PAGA is insufficient or inadaptable to perform their analysis.

Wolf, F.A., Hamey, F.K., Plass, M., Solana, J., Dahlin, J.S., Göttgens, B., Rajewsky, N., Simon, L., and Theis, F.J. (2019). PAGA: graph abstraction reconciles clustering with trajectory inference through a topology preserving map of single cells. *Genome Biol.* 20, 59.

Response: We have experience working with different pseudo-temporal analyses such as Monocle3, URD, and PAGA. All of these perform well under the circumstance where the cell-cell network, as well as clustering, are primarily driven by differentiated cell types across all stages. However, in our study, the networks of earlier time points (1.5, 2 dpf) are largely driven by regional patterning information, with accessible chromatin for later differentiation not driving cell clustering. Hence, in PAGA analysis we lose this minor chromatin accessibility information prefiguring later differentiation. To overcome this, we developed a new constellation type of analysis to retrogradely map back chromatin accessibility from later differentiated cell type stages onto earlier regional patterning stages, *irrespective of the chromatin*

accessibility driving cell clustering at these earlier stages. In our hands, only constellation analysis and not PAGA allows us to reveal the earliest opening of chromatin prefiguring later differentiated cell types.

We now better explain the rationale for the Constellation analysis over approaches such as PAGA.

Line 253-256: “To understand when cluster-specific peaks become established, as well as cluster relatedness, we developed the bioinformatics pipeline “Constellations” to retrogradely map accessible chromatin regions related to differentiated cell types onto earlier stages preceding overt differentiation.”

1.4 - Regarding lines 87-90, a supplementary table should be prepared which describes in detail the fractions of cells which were collected from the FACS, the number of cells loaded per 10x analysis, the number of cells retained after thresholding per library, the median genes per cell per library, and the actual read depth per library. Similar relevant data should be provided to support the snATACseq datasets. These technical statistics will build confidence in the generation of the author’s datasets as well as ensure the reproducibility and interpretability of this study in the future. Additionally, the number of animals used per time point collected per replicate, as well as information as to the sex of older animals should be reported, possibly to also be included in this table.

Response: We now include all the requested information in a new **Supplementary Table S1**.

1.5 - Surprisingly, there is no discussion or search for chromatin remodeling factors anywhere in the results. This analysis would greatly enhance the strength of the conclusions and should be included in some form. If not, if the authors could explain why, that would be helpful.

Response: We now show expression of chromatin remodelers across stages and major progenitor clusters in a new Supplementary Figure 17. While we do not have space to discuss the many chromatin remodelers in this manuscript, we hope this information will be of use for future studies aimed at uncovering the mechanistic basis of opening lineage-specific neural crest enhancers.

Lines 189-191: “We observe the largest enrichment of chromatin remodelers in the ectomesenchyme population at 1.5 and 2 dpf and tissue-specific progenitors at 3 and 5 dpf, suggestive of large-scale remodeling of lineage-specific enhancers at these stages.”

1.6 - At line 94, A table should be included clearly listing the specific marker genes utilized to categorize each cluster in each dataset as well as appropriate citations to support each gene’s use. Current reporting of total genes listed per cluster in table 1 is valuable and should not be removed.

Response: In new Table S3, we now list specific marker genes for each cluster, as well as citations or references to in situ validation within our paper.

1.7 - At line 392 (referenced at line 83-84), the specific boundaries utilized to define the cranial domain collected per time point should be included.

Response: We have now modified the Methods to describe how the heads were dissected.

Lines 470-472: “Heads from converted *Sox10:Cre; actab2:loxP-BFP-STOP-loxP-dsRed* fish were decapitated at the level of the pectoral fin at all stages (1.5 dpf - 210 dpf). Eyes and brains were removed (eyes in 3 dpf-210 dpf, brains in 60 dpf – 210 dpf).”

1.8 - The authors assume that the number of clusters demonstrated in their 14 dpf dataset, which underpins much of their analysis, is representative of 23 distinct tissue classes as they identified 23 clusters. However, the number of clusters will robustly shift depending on the resolution parameter set during the analysis. This assumption may impact the degree of sensitivity of this analysis, by which many conclusions in the paper are drawn. Rationale is required for the arbitrary setting of the 0.8 resolution parameter by which the clustering was formed. Further, the robustness of their novel constellation analysis should be demonstrated at a range of resolution parameters. Alternatively, if the authors have data demonstrating definitively that 23 separate tissue lineages are present at 14 dpf would also support their case.

Response: We agree that cluster number determination is a somewhat arbitrary process. Indeed, in our analysis we identify multiple flavors of the “same” cell type, for example “stroma 1” and “stroma 2”. We have therefore focused on showing that our Constellations analysis is robust to changes in cluster number and resolution. In new Fig. S21, we show that the five major groupings of cell types are robust to changes in resolution from 0.4 to 1.4 and number of clusters from 14 to 24. We also provide more details in the Methods of how

Line 263-265: “The prediction of major cell groupings by Constellation analysis is robust to changes in resolution, which alters the number of predicted clusters at 14 dpf (Fig. S21a-e), as well as to inclusion of later stages (60 and 210 dpf) (Fig. S21f).”

1.9 - At line 235, regarding the establishment of the constellation analysis, it is not clear why the authors cut off their analysis at 14 dpf when additional datasets at 60 dpf and 150 dpf were also at hand. Validation of the results from the constellation pipeline including these datasets would be a dramatic improvement to the current version of the analysis.

Response: In a new Supplementary figure 21, we now show Constellations analysis using two later timepoints (60 and 210 dpf) and get similar results to what we had previously shown without these stages. However, we have maintained the original Constellations analysis out to 14 dpf in main Figures 4 and 5. Our goal was to use chromatin signatures at 14 dpf to map back the origin of chromatin pre-pattern to earlier stages. As all the major cell types are specified by 14 dpf, we find that starting the retrograde analysis at this stage gives the clearest and most informative analysis. One reason for this is that we find that enhancers continue to change during aging, with these new aging-specific enhancers at later adult stages adding noise to the Constellations analysis. We have therefore only included the later timepoints in the new supplementary figure to make the point that their inclusion doesn't change the fundamental interpretations of the data.

1.10 - In somewhat of a continuation of the prior comment, at line 105, the authors claim to find “17 distinct clusters” following their analysis in Seurat. However, no discussion has been offered about how clusters were determined to be accurate, undermining this claim. An explanation as to how the authors arrived at the appropriate number of clusters using the 0.8 resolution parameter is needed.

Response: As described in **1.8**, we agree that the exact number of clusters is somewhat arbitrary. We have now confirmed the dataset was not over-clustered using the 0.8 resolution by performing the subsampling-based approach “chooseR” as described in Patterson-Cross et al., 2021. We also ensured that all known cell types (e.g. cartilage, bone) were represented, and we note that we uncovered multiple flavors of several cell types (e.g. “stroma 1” and “stroma 2”) that might reflect some retention of regional patterning information from earlier stages. One reason the adult stage may have 17 clusters versus 23 clusters at 14 dpf is due to collapse of related clusters into a single one (e.g. a single “stroma”), as well as some progenitor populations disappearing (e.g. we do not recover “perichondrium” after 14 dpf).

1.11 - The conclusions drawn from the motif enrichment analysis discussed in lines 268-278 is a very light treatment, failing to reflect much of the nuance of expression of the transcription factors in multiple

lineages. For example, the ETS2 motif is only addressed in the context of the pillar cells, while it clearly is also expressed in the stromal constellation as well. Additionally LEF1 is characterized within the Dermal constellation when there is involvement in the skeletogenic and gill constellations. In total, the authors have truly uncovered a combinatorial code of these transcription factors paired with specific open chromatin motifs. This astonishing achievement would be better highlighted in this text, rather than relying on such a heuristic approach to the interpretation of these graphs.

Response: We have modified the text to emphasize that our study suggests unique combinatorial codes for lineage-specific transcription factors.

Lines 299-305: “However, these individual factors are not exclusively associated with particular cell types, and hence it is likely particular combinations of factors that drive lineage commitment (Table S6). For example, ETS2 and GATA3 show strongest co-enrichment in the pillar cell trajectory, suggesting they could influence development of this unique type of gill-associated endothelial cell. These findings therefore show the power of Constellations analysis to reveal potential combinations of factors for establishing regional chromatin accessibility important for later cell type differentiation.”

1.12 - I am confused how approximately 50% of the merged scRNAseq atlas that is present in Figure 2a can be unidentified when there are assigned lineage identities to every population that is present in the individual datasets presented in the supporting supplemental figures. Further, labels applied in the supplements do not correspond to the labels used to describe lineages in the atlas. The authors should endeavor to make these labels congruent to help facilitate the reader’s ability to see the source for each cell lineages shown in the merged STITCH plot.

Response: As the STITCH plot focuses on the emergence of differentiated cell types, undifferentiated mesenchyme was labeled as “unspecified” in grey. In contrast, we assigned a name to each UMAP cluster in the individual datasets (i.e. both undifferentiated mesenchyme and differentiated cell type clusters) to the best of our abilities using known marker genes as well as our in situ validation (now described with references in new Table S3). While we can certainly identify areas of the STITCH plot (currently labeled “unspecified”) that correspond, for example, to dorsal, ventral, anterior, and posterior early arch domains, we used the STITCH plot primarily to highlight differentiated cell types at later stages. In the text, we now clarify that regions of the STITCH plot were assigned cell type identities based on the in situ validation shown in the supplementary figures.

Lines 134-137: “We then used in situ validation to assign regions of the scRNAseq STITCH plot corresponding to major cell types, and then transferred identities to the snATACseq STITCH plot based on chromatic accessibility across gene bodies (i.e. “pseudo-transcriptome”) (Fig. S9, S10).”

We have also now ensured that cluster names in UMAP and STITCH are congruent.

1.13 - Paragraph from line 216-230 is a superb treatment of characterizing the findings of the SnapATAC data within the context of the literature, but fails to leverage the true power of this dataset to identify novel or unexpected motifs. Please include some analysis or discussion on this matter.

Response: While space limitations do not allow us to discuss the breadth of motifs discovered, we do discuss a novel NR5A2 motif associated with the ventral domain of the mandibular arch. In ongoing work, we have a manuscript in preparation describing in detail roles of this unexpected orphan nuclear receptor in patterning the mandible. We have also modified the concluding sentence of this paragraph to emphasize that both known and novel motifs have been discovered and are listed in Table S1.

Lines 243-246: “This approach shows the power of integrated scRNAseq and snATACseq data to predict the spatial expression domains of the vast majority of CNCC ectomesenchyme genes at pharyngeal arch stages, as well as in uncovering both known and novel domain-specific transcription factor binding motifs such as NR5A2 (Table S2).”

1.14 - Beginning at line 517, the definition of equations is confusing and circuitous. Standardized equation should be used when possible. Further, at line 520, the authors define $D(\text{tissue})$ with text rather than through a standardized equation format. A similar issue occurs at line 522. Lastly, I believe the authors intended to define a as 8 at line 525, though it is unclear. Care should be taken to revise this paragraph for clarity as it is essential to their Constellation method.

Response: We have updated the corresponding Methods section with standardized equations for more accurate description of our pipeline. We now also clarify that $a=12$ (in carefully revising this section we realized the value should be 12 and not 8).

1.15 - At line 129, the Authors claim that their evidence is clearly apparent for snATACseq, but then explain all their findings using the scRNAseq plots without clarifying this point. Please address this discrepancy or change the claim to reflect the scRNAseq.

Response: In Fig. 2a,b, we show that major skeletogenic, gill, dermal fibroblast, stromal, and tooth cell types form discrete groupings in both scRNAseq and snATACseq STITCH plots. We now better clarify this in the text.

Lines 134-145: "We then used in situ validation to assign regions of the scRNAseq STITCH plot corresponding to major cell types, and then transferred identities to the snATACseq STITCH plot based on chromatic accessibility across gene bodies (i.e. "pseudo-transcriptome") (Fig. S9, S10). As early as 3 dpf (particularly apparent for snATACseq), we observe partitioning of CNCCs into skeletogenic versus gill lineages. Skeletal-associated regions include *hyal4+* perichondrium, periosteum, tendon/ligament cells, chondrocytes, and osteoblasts (Fig. S9), and gill-associated clusters include an *fgf10b+* gill progenitor population, gill chondrocytes, pillar cells, and tunica media cells (Fig. S10). Starting at 3 dpf, dermal fibroblasts form a separate branch in both scRNAseq and snATACseq STITCH plots, and *cxcl12a+* stromal cells and teeth are closely associated with less differentiated mesenchyme in the center of the plots (Fig. 2a,b; Fig. S9, Fig. S11)."

1.16 - The authors claim at line 132 that the *fgf10b+* population is present at 5 dpf and direct the reader to Figure S10, but do not show data relevant to this time point. Please include micrographs reflecting these findings in 5 dpf animals.

Response: In new Supplementary Figure 10g, we now include in situ hybridization for *fgf10b* at 14 dpf that shows expression in the tip of the gill filament, similar to what we had previously showed at 30 dpf (Supplementary Figure 10e). In situ hybridization at 5 dpf is quite difficult, as it is difficult to find the very early gill buds in sections and this stage is too late for robust whole-mount in situ hybridization. Instead, we now note the expression of the *fgf10b:nlsEOS* knock-in transgene in the gill filaments at 6 dpf (Supplementary Figure 14d).

1.17 - Same comment above, regarding data discussed in lines 159-162, please show 5 dpf/14 dpf animals.

Response: In new Supplementary Figure 10g, we use double RNAscope ISH at 14 dpf to show expression of *rdh10a* in the filament base, opposite to expression of *fgf10b* in the filament tip. We also now show expression of *cxcl12a* in the filament base at 14 dpf.

1.18 - Line 134-135 the claim that no cardiomyocytes is found is very misleading as the population of NCC sampled should not include these cells. Further, the supporting figure does not support this claim. Please strike this clause.

Response: We have deleted this clause.

1.19 - At line 140, it is difficult to interpret the lack of expression within datasets not fully knowing the read depth of each dataset and given the relatively low genes/cell value reported. How are the authors sure that this inconsistency in their dataset with prior literature, which they cite, not point to a more systemic issue with the resolution of their datasets?

Response: We now add the word “transient” to clarify that the reacquisition of pluripotency as recently reported by the Wysocka group was extinguished before the post-migratory CNCC stages we are studying. We also qualify that we cannot rule out low level expression of pluripotency factors, e.g. gene drop-out due to sequencing depth. However, as pluripotency is typically defined by very high co-expression of these factors, our results strongly argue against similarly high co-expression of these factors in post-migratory ectomesenchyme.

Lines 150-154: “Although formation of CNCC ectomesenchyme involves a transient reacquisition of the pluripotency network¹⁴, we also did not observe expression of pluripotency genes *pou5f3* (*oct4*), *sox2*, *nanog*, and *klf4* at any stage of post-migratory ectomesenchyme, although we cannot rule out low expression of some or all of these genes.”

1.20 - Line 152, the authors claim that there are two distinct lineages, one of which is defined by *cxcl12a+/ccl25b+* and the other by solely by *hyal4+* cells, however there is clearly a *hyal4+/cxcl12a+* domain shown in figure 2e. Please address this discrepancy.

Response: We now clarify that *hyal4+* cells express lower levels of *cxcl12a* than the stromal cells.

Lines 161-164: “For skeletogenic clusters, cell distribution from 5 to 14 dpf suggested two distinct lineages: one involving chemokine-expressing stromal cells (*cxcl12a+/ccl25b+*) and a second emanating from *hyal4+* cells that express lower levels of *cxcl12a* (Fig. 2c-e, Fig. S13).”

Minor Issues:

1. At line 67, the authors state that “These [prior] findings point to extrinsic inductive cues of CNCC fate determination,” which does not seem to be in line with the hypothesis they propose on line 68. Perhaps a clause is missing to connect the two thoughts?

Response: We have added language to connect these two thoughts. We have also toned down our discussion of extrinsic cues in response to Reviewer #2.

Lines 69-76: “And in skate, mesodermal cells can contribute to gill cartilage, a classically considered CNCC-derived structure, suggesting that, rather than cell potential being a unique intrinsic property of CNCCs, extrinsic cues from the local arch environment may induce similar cell fates in mesenchyme of diverse origins in certain contexts⁹. Here we take a genomics approach in zebrafish to understand when enhancers linked to diverse CNCC fates first gain accessibility. In so doing, we find that chromatin accessibility underlying multilineage potential is largely gained after CNCC migration, arguing against an epigenetic pre-pattern in premigratory CNCCs for diverse fate potential.”

2. The authors do a remarkable job presenting a vast amount of data in a short span. Summary statements at the end of each major section would make this dense body of work more accessible.

Response: Where missing, we have added new summary statements to the end of each major section.

3. In Figure 2i, the arrows do not point to cells that are new EOS+ cells, but rather to cells which were previously converted and have since continued to express *fgf10b*. On line 170, the authors state that the cells at the new pillar cells should be “green only,” however cells shown in Figure 2i, at day 10 are clearly green and magenta. Either the text needs to be corrected to reflect the anticipated co-labeling of the cells at the tips or a new yellow arrow needs to be drawn to point to the intended population of cells.

Response: Yes the yellow arrow indicates converted pillar cells that continue to express new green *fgf10b:nEOS*. We have deleted the phrase about new EOS+ cells being added to make it clearer that converted EOS+ cells contribute to both pillar cells and gill chondrocytes.

4. At line 108-109, there is a redundant usage of “and” in the list of lineage derivatives.

Response: This has been fixed.

5. In supplemental figures in which a single letter is used to designate a group of images, the inclusion of some sort of boundary (eg. a thin boxed outline or shaded background) to designate which panels specifically belong to that letter designation would greatly improve the figure clarity.

Response: We now better group panels for single letter designations by adding extra space between different groupings.

6. At line 110-112, the authors claim that this finding is “unexpected,” but do not offer an explanation as to why. Please include a clause to make this point clear.

Response: We deleted “unexpectedly”.

7. At line 116, citations are needed to support the use of these marker genes. May also be resolved through the inclusion of the genes in the marker gene table requested in the major issues section.

Response: We include citations and marker genes in new Table S3.

8. At line 148-149, the authors use specific gene combinations to define posterior arch mesenchyme but defer generally to a “skeletogenic” signature for their anterior arch mesenchyme. Please define the set of genes used to identify “skeletogenic” cells.

Response: We have changed “skeletogenic” to “jaws” to better reflect that this refers to anterior arch mesenchyme derivatives. We have also modified the Methods to better describe how “gills and “jaws” populations were defined.

Lines 525-528: “After removing of non-mesenchyme derived populations as well as dermal fibroblast (*pah+*) and teeth (*spock3+*) populations, we placed *hoxb3a+/gata3+* cells into a “gill” cluster and all other cells into a “jaws” cluster.”

9. Equilateral triangles used as arrow heads obscure the direction intended to be seen in some micrographs. Change in arrowhead shape or the additional of tails will clarify meaning. Different arrow shapes are also used throughout the paper, which detracts from the style.

Response: We have changed arrowheads to arrows (except for Fig. 6e where the arrowhead is not equilateral) and made arrow styles consistent.

10. At line 152, in figure 2C-E, there is a strange aberrance in the way the pseudotime graph was learned. It appears that the presumptively more differentiated cell population at 14 dpf in the lower half of the gill stroma (Fig. S13e, red population) is leading to a 5 dpf population which is flagged as earlier according to the color scale. Please verify the correct branch node was used to establish the graph learning for Figure 2d.

Response: We thank the reviewer for catching this inconsistency. We have now redone the pseudotime analysis using a starting node within the 5 dpf stroma population.

11. Line 162, I am not convinced by these data alone that the *cxcl12a+/ccl25b+* cells are not also contributing to the perivascular population, which is also proximal to the basal root of the pseudotime tree. Can you provide further support to address this possibility?

Response: We agree that the origins of tunica media and perivascular cells are unresolved, especially as due to technical reasons we were unable to test the contribution of *fgf10b:nEOS* cells to these populations. We have reworded this section accordingly.

Lines 200-202: "These data support *fgf10b+* cells being progenitors for gill cartilage and pillar cells from larval through adult stages; the origins of tunica media and perivascular populations remain unresolved."

12. In Supplementary Figure 10d, the white text is nearly impossible to read. Please change color of text and micrograph label to improve visibility.

Response: We have updated this text to be just plain black.

13. Line 167, please replace "over time" with a specific interval of time you assayed.

Response: We clarify that this is "from 6 dpf to 2 years of age".

14. Around Line 465, please identify the programming environment used for this analysis as well as the version number if different from R.

Response: On Line 503 we now state that we used MATLAB (R2018b).

15. Please clearly label UMAPS on black backgrounds to distinguish them easily from microscopy images.

Response: "UMAP" labels have been added to Fig. 3, Fig. 5, Fig. S18.

16. In supplementary figure 11, 12, and 13, cell type labels above the STITCH graphs along with the gene name shown would greatly enhance the accessibility of these data, similar to the format used in Figure supplement 10.

Response: We have added cluster names next to the marker genes for the STITCH graphs in Supplementary Figure 13. As Supplementary Figure 11 refers entirely to dermal fibroblast markers and Supplementary Figure 12 to multipotency genes (not cell types), we did not add additional cluster names over STITCH graphs for these figures.

17. Several paragraphs begin with the same formulaic introduction “To XXX, we YYY.,” which undermined the exciting data contained within. Improvements in the writing style would make the flow of the paper more pleasant to read.

Response: We have edited the writing style to reduce repeat of the “To XXX, we YYY” formula and to more generally make the paper easier to read.

18. Starting at line 201 to the end of that paragraph, this speculation, while intriguing, should be placed in the discussion as it is not a relevant result.

Response: We have moved this to the Discussion.

19. Several intriguing speculations are presented throughout the study. Due to the discovery of phenylalanine and tyrosine catabolic pathways in CNCC-derived fibroblasts, the authors posit that these pathways may drive the etiology of Alkaptonuria. Further, the authors speculate that cxcl12a+ populations in zebrafish contribute to skeletal regeneration, unlike in mouse models. Both of these speculations would be better suited for in the discussion rather than the results.

Response: We have moved all speculation related to Alkaptonuria and cxcl12a+ populations to the Discussion as requested.

20. A commitment should be made in the Data Availability section at line 553 to provide access to the code used in the generation of this project.

Response: We had added a Code Availability section and have uploaded the essential R codes for recapitulating our results at Github (https://github.com/TedKCT/Crump_zebrafish_scNCC).

21. In Supplement figure 14, the number animals differs from the number of animals reported in line 388. Please edit to reflect.

Response: We have corrected this error in the text to reflect the accurate n numbers.

22. At line 406, please provide justification as to why 126 cycles was selected for this sequencing while only 65 cycles was selected at line 417 for the ATAC sequencing.

Response: We now explain that for scRNAseq “Read2 was extended from 98 cycles, per manufacture’s instruction, to 126 cycles for higher coverage”, and that for snATACseq “Both read1 and read2 were extended from 50 cycles, per manufacture’s instruction, to 65 cycles for higher coverage.” In general, RNAseq and ATACseq experiments require their own optimized amplification parameters.

23. At line 453, if a work flow was utilized in the Seurat package to integrate the datasets, please explain which workflow was used.

Response: We now explain that we used “Seurat functions JackStrawPlot() and ElbowPlot()”.

24. Please include information about code generation and data interpretation in the author statement.

Response: These have been added to the Author Contributions section.

REVIEWER #2:

2.1 - The use of Sox10CRE +ve cells ensures that all the cells analyzed are NCCs (with the exception of a few otic vesicle cells where there is also expression). This makes for a clean study, but it leaves me confused about the overarching conclusion of the importance of extrinsic factors. Specifically, I find that while the data are compelling and well presented, the intellectual framework —both the rationale for the study and the overarching conclusions lacks equivalent clarity. The authors essentially set up a dichotomy between intrinsic and extrinsic patterning at the onset of the paper, but do not lay out what experimental outcomes would support or refute each of these models. Moreover, whether intrinsic patterning is equivalent to the concept of “pre-patterning” is not discussed, and I would certainly appreciate citation of the chick transplant work of Drew Noden (1983) that led to the general idea of pre-patterning in the CNC cell population. Importantly, the term intrinsic can be used at either a single cell level or at a tissue level, and the authors do not make their meaning clear. However, as their approach focuses exclusively (and appropriately) on the CNCC lineage it obviously has less to say about the role of external tissues. Nevertheless, in either case, ‘extrinsic’ implies a role for adjacent cells and the secreted – or possibly membrane bound – molecules from those adjacent cell types that influence the cell type of interest. Thus, an important if indirect approach to recognizing extrinsic activity is a requirement for specific receptor function. While Wnt5 and Fgfa10 signals are touched upon, this overall idea is not well discussed.

Response: We agree that our study does not explicitly address extrinsic versus intrinsic patterning of CNCC derivatives, but rather the timing at which enhancers linked to diverse neural crest fates first become accessible. We have therefore substantially removed/revised discussion of extrinsic factors and now clarify that our study shows that lineage-specific enhancers are largely opened after CNCC migration into the arches. We have also added in the Noden (1983) reference, as we agree that this demonstrates some of the earliest evidence for a pre-pattern in CNCCs, though we have also added in subsequent work from Couly and LeDouarin showing a major role for endodermal epithelia in patterning the facial skeletal elements. We also note that our study provides a rich resource for understanding potential receptors in CNCCs that could respond to extrinsic signals, but we do not discuss this in depth as we feel it is well beyond the scope of the current study.

Abstract Lines 47-49: “Rather than multilineage potential being established during cranial neural crest specification, our findings support progressive and region-specific chromatin remodeling underlying acquisition of diverse potential.”

Lines 62-64: “The extent to which diverse lineage potential is pre-patterned in CNCCs, or acquired after their migration to the arches, has been investigated for over a century through labeling, grafting, and extirpation experiments³.”

Lines 66-71: “In addition, grafting of avian CNCCs between arches showed that they retain a pre-pattern of which skeletal elements to make, such as the beak⁵. On the other hand, grafting of endodermal epithelia can induce supernumerary skeletal elements of morphologies corresponding to the location from which epithelia are taken, suggesting an important role of external cues in CNCC fate⁶.”

Lines 79-81: “...we find that chromatin accessibility underlying multilineage potential is largely gained after CNCC migration, arguing against an epigenetic pre-pattern in premigratory CNCCs for diverse fate potential.”

Lines 370-373: “Rather than lineage-specific chromatin accessibility being established during CNCC specification, our Constellations analysis points to the progressive remodeling of chromatin underlying diverse cell type differentiation.”

2.2 - Adding to the confusion are brief allusions to the work of the Gillis lab demonstrating that mesoderm-derived cells contribute to the skate gill skeleton. In the Discussion the author's comments appear to be hinting that these mesoderm-derived cells may come under a common influence together with adjacent CNCCs. If this is the authors' meaning then it needs to be spelled out, and note made that this is speculative rather than based on data. In the Introduction the same work from Gillis is cited as part of the rationale for the study, with the authors stating that these findings "point to extrinsic inductive cues for CNCC fate determination". At least as currently written I find this logic opaque.

Response: We apologize this was not made clearer and have clarified discussion of this work in the Introduction and removed it from the Discussion. We cite this study as it shows that mesoderm and CNCC lineage cells have equal capacity to form cartilage in the gills of skate, arguing against CNCCs having a unique pre-pattern for this classically considered CNCC derivative.

Lines 73-77: "And in skate, mesodermal cells can contribute to gill cartilage, a classically considered CNCC-derived structure, suggesting that, rather than cell potential being a unique intrinsic property of CNCCs, extrinsic cues from the local arch environment may induce similar cell fates in mesenchyme of diverse origins in certain contexts⁹."

REVIEWER #3:

3.1 - Data suggest that *fgf10+* cells are the origin of 4 specialized gill subtypes. They generate *fgf10+:eos* line and use it follow the fate of cells that expressed *fgf10+*. They discover that the lineage label ends up in two of the 4 subtypes (gill cartilage and pillar cells). What about the other two gill subtypes – tunica media and perivascular populations? Some comment is merited here.

Response: We have revised the Results to indicate that our data only show contributions to gill cartilage and pillar cells. The red fluorescence signal after *fgf10b:nlsEOS* photoconversion does not survive fixation, and hence we cannot use RNA or protein probes to detect cell types. For the *fgf10b:nlsEOS* work, we were able to use the distinctive cell morphology of chondrocytes (large cell body surrounded by abundant matrix) to identify these cells in live photoconverted animals, and likewise pillar cells have a distinctive cell morphology and are the main CNCC-derived cell type in the secondary filaments. However, tunica media and perivascular cells have a generic fibroblast-like morphology and in our hands can only be identified by marker expression (e.g. the RNAscope in situ pictures we present in Fig. S10). We were therefore unable to determine whether these are also derived from *fgf10b+* cells (or alternatively *cxcl12a+* cells as reviewer 1 suggests).

Lines 200-202: "These data support *fgf10b+* cells being progenitors for gill cartilage and pillar cells from larval through adult stages; the origins of tunica media and perivascular populations remain unresolved."

3.2 - I felt that the manuscript suffered a bit from the style of the writing. Stronger topic sentences that emphasized the main conclusion presented in each paragraph would make the central points of the study easier to understand. When I first read it I felt that I had learned a little bit about a lot of things.

Response: We have added additional topic sentences to the beginning of each Results sub-section, as well as summary statements at the end of these same sections.

Reviewers' Comments:

Reviewer #1:

Remarks to the Author:

Congrats on accomplishing this important study!

Reviewer #2:

Remarks to the Author:

The manuscript has been much improved by the extensive revisions, and all my concerns have been appropriately addressed.